# Differentially-private and plausible counterfactuals

## Abstract

Counterfactual explanations are particularly appealing in high-stakes domains such as finance and hiring, as they provide affected users with suggestions on how to alter their profiles to receive a favorable outcome. However, existing methods are characterized by a privacy-quality trade-off. More precisely, as highlighted in recent works, instance-based approaches generate plausible counterfactuals but are vulnerable to privacy attacks, while perturbation-based methods offer better privacy at the cost of lower explanation quality. In this paper, we propose to solve this dilemma by introducing a diverse set of differentially-private mechanisms for generating counterfactuals, providing high resistance against privacy attacks while maintaining high utility. These mechanisms can be integrated at different stages of the counterfactual generation pipeline (*i.e.*, pre-processing, in-processing or post-processing), thereby offering maximal flexibility during the design for the model provider. We have performed an empirical evaluation of the proposed approaches on a wide range of datasets and models to evaluate their effects on the privacy and utility of the generated counterfactuals. Overall, the results obtained demonstrate that in-processing methods significantly reduce the success rate of privacy attacks while moderately impacting the quality of counterfactuals generated. In contrast, pre-processing and post-processing mechanisms achieve a higher level of privacy but at a greater cost in terms of utility, thus being more suitable for scenarios in which privacy is paramount.

## 1 Introduction

The use of machine learning models has become widespread in many spheres of our society. However, explaining how these models work is essential for fostering trust in their outcomes (Guidotti et al., 2018; Molnar, 2020; Vashney, 2022). In particular, in this work, we focus on counterfactuals (Wachter et al., 2017), which are a form of explanation that suggests modifications to the input profile aiming to alter the model's prediction (Wachter et al., 2017). As such, counterfactuals are one of the popular post-hoc techniques to provide explanations to affected users to help them improve their predictions (Hamer et al., 2024). However, recent works have shown that an adversary can also exploit the information provided by counterfactuals to conduct privacy attacks. For instance, counterfactuals can be leveraged to perform membership inference and model extraction attacks (Shokri et al., 2020; Aïvodji et al., 2020; Dissanayake & Dutta, 2024). Thus, it is important to investigate the privacy risks associated with counterfactuals as well as to develop countermeasures to mitigate them.

One popular category of methods for generating counterfactuals is perturbation-based approaches (Wachter et al., 2017), which generate counterfactuals by perturbing the feature values of an instance towards the decision boundary. However, they have been criticized for their lack of plausibility (Laugel et al., 2019). To address these limitations, researchers have proposed instance-based counterfactual methods (Laugel et al., 2018; Keane & Smyth, 2020), which use samples from the training dataset as a basis for counterfactual generation. A representative method of this family is NICE (*Nearest Instance Counterfactual Explanation*) (Brughmans et al., 2023), which selects the nearest instance in the training set to generate a counterfactual. However, as shown by Goethals et al. (2023), this type of counterfactual is vulnerable to explanation linkage attacks (Dunn, 1946) due to their use of training instances. In a nutshell, this attack links the adversary's existing knowledge with counterfactual data to extract sensitive information about profiles from the training

set. The authors also proposed a $k$-anonymization approach to mitigate this vulnerability. However, $k$-anonymity (Samarati & Sweeney, 1998) is known to be vulnerable to homogeneity attacks (Machanavajjhala et al., 2007), when all instances in one $k$-anonymized group share the same sensitive attribute values.

In our setting, the training dataset is considered private and users are assumed to have black-box access to the model (more precisely, label-only), meaning that when they query it, they only receive the model's prediction regarding their query. In addition, they also receive as explanation one counterfactual specific to their query (*e.g.*, if the model's decision is considered undesirable). This generated counterfactual is not considered to be public information and is typically only available to the user who submitted the query.

*Summary of contributions.* Given that perturbation-based counterfactuals often suffer from low plausibility and instance-based counterfactuals are vulnerable to explanation linkage attacks, we propose a hybrid approach that leverages the strengths of both worlds. Specifically, our framework uses instance-based counterfactuals to improve plausibility while incorporating differential privacy (DP) guarantees (Dwork et al., 2006) to prevent explanation linkage attacks. In addition, our framework is flexible enough to enable differential privacy at various steps of the counterfactual generation pipeline, as illustrated in Figure 1.

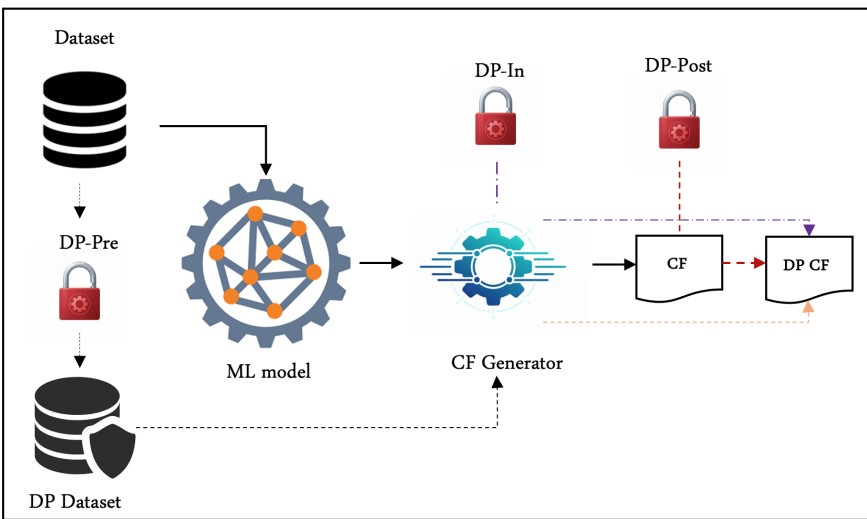

Figure 1: Illustration of the different steps of the counterfactual generation pipeline in which our mechanisms for generating differentially private counterfactuals can be applied.

More precisely, our main contributions can be summarized as follows:

- **Pre-processed differentially private counterfactuals (`DP-Pre`).** In this method, we first generate differentially private data using the Laplace mechanism (Dwork et al., 2014) and randomized response (Warner, 1965). This differentially-private data is then used as input to the instance-based counterfactual generation mechanism. This approach ensures that the input data itself is differentially-private, mitigating privacy risks at the initial data processing stage without requiring any changes to the rest of the counterfactual generation pipeline.

- **In-processed differentially private-counterfactuals (`DP-In`).** We have also designed algorithms to implement differential privacy directly in the counterfactual generation mechanism. More precisely, `Inline_DP` is a NICE-based counterfactual approach that works by selecting $k$ nearest neighbours and then applying the Laplace mechanism (Dwork et al., 2006) and randomized response (Warner, 1965) to the feature values to satisfy differential privacy. Afterwards, the counterfactual is generated based on these values. This approach ensures that the features are perturbed directly, thus protecting privacy during the counterfactual generation process. This in-processed mechanism was specifically designed to be integrated into `NICE`, although it can be adapted to other instance-based counterfactual generation methods.

- **Post-processed differentially private counterfactuals (`DP-Post`).** In this approach, we apply the randomized response (Warner, 1965) and the Laplace mechanisms (Dwork et al., 2006) as a post-processing[1] step on the counterfactuals generated by the instance-based method. This approach provides a robust privacy-preserving layer after the initial generation of instance-based counterfactuals.

To assess their performance, we evaluated the differentially-private counterfactual generation methods across multiple datasets and privacy regimes, using diverse metrics such as counterfactual plausibility and re-identification rate. The results obtained demonstrate their effectiveness in mitigating explanation linkage attacks. We have chosen `NICE` as the instance-based counterfactual generation method because it outperforms other instance-based methods, such as CBR (Keane & Smyth, 2020) and SEDC (Fernández-Loría et al., 2020) and perturbation-based methods like DiCE (Mothilal et al., 2020) and CFproto (Van Looveren & Klaise, 2021) as shown in (Brughmans et al., 2023). However, it is possible to directly apply our pre-processed and post-processed techniques to any other instance-based counterfactual approach, while our in-processed mechanisms may require some adaptation regarding how they choose their basic instance and generate counterfactuals.

*Outline.* The paper is organized as follows. First, in Section 2, we introduce the background notions of counterfactuals and differential privacy necessary for the understanding of our work before describing the existing related work in Section 3. Afterwards, we present our proposed solutions in Section 4. Finally, in Section 5, we present the experimental findings and analysis to assess the efficiency of each solution in mitigating explanation linkage attacks before concluding with future works in Section 6.

## 2 Background

In this section, we introduce the background on counterfactuals as well as differential privacy.

### 2.1 Counterfactuals

As highlighted by Martens & Provost (2014) and Wachter et al. (2017), counterfactual explanations differ from other forms of post-hoc explanation techniques in that they suggest changes to the input profile to alter the model's prediction. More precisely, given an input profile with feature values $x_1^0, \ldots, x_1^d$ and the corresponding model's prediction $y_1$, a counterfactual explanation method generates a counterfactual with feature values $cf^1, \ldots, cf^n$ satisfying two conditions : (1) The model should assign it a different prediction than the original instance and (2) it should be close to the original instance in terms of a predefined distance, with the Euclidean distance being one of the most commonly used in counterfactuals. This means that counterfactual explanations should be generated by making as few changes as possible to the input profile features, yet still resulting in a different prediction. Various metrics have been defined to measure the quality of counterfactuals (Karimi et al., 2022). For instance, `Proximity` measures the distance between the counterfactual and the original instance while `Sparsity` refers to the minimum number of features that need to be changed to achieve the counterfactual. Finally, `Plausibility` measures if the counterfactuals generated are realistic with respect to the training data distribution. For example, consider a bank customer who applies for a credit limit increase and receives a rejection. She may wish to understand how to modify her financial profile to obtain approval or compare her case with similar applicants who were successful, to understand the reasons for the denial. To meet the aforementioned criteria, the counterfactual profile generated should be relatively similar to the original instance (proximity), involve only minimal changes to the original features (sparsity), while remaining within the bounds of realistic scenarios (plausibility). Thus, it would be unrealistic to suggest that the applicant should earn a Ph.D. by the age of eighteen.

While counterfactuals can be generated in theory for any data types including image (Goyal et al., 2019; Delaney et al., 2023; Mishra et al., 2024) and text (de Oliveira et al., 2024), in this work we focus on tabular datasets. The two main families of approaches for counterfactual generation are perturbation-based

---

[1]Note that we refer to "post-processing" as the application of differentially-private mechanisms to counterfactuals after their generation, rather than the broader definition of post-processing in the context of differential privacy.

and instance-based algorithms (Karimi et al., 2022). Perturbation-based algorithms adjust feature values towards the decision boundary to achieve the desired model response. While these algorithms minimize the distance between the original instance and counterfactuals, they lack plausibility because they apply the changes towards the decision boundary without considering how these changes affect the generated profile. Instead, instance-based algorithms select one instance from the training dataset's counterfactual class and use it to generate a counterfactual. The exact process to derive counterfactuals from this instance varies among existing algorithms. In this work, we rely on `NICE` (Brughmans et al., 2023), which is one of the state-of-the-art instance-based algorithms that improve on proximity and plausibility in comparison to other instance-based counterfactuals. In a nutshell, `NICE` first identifies the nearest neighbour of the original instance for which the model makes a different prediction. Then, through an iterative process, the feature values of the considered instance are replaced with the values of the nearest neighbour until the model changes its prediction. The selection of the feature values is based on a reward function, which integrates several criteria related to the quality of counterfactuals, namely proximity, sparsity and plausibility.

## 2.2 Differential Privacy

Differential privacy (DP) is a privacy model that provides strong guarantees by ensuring that the contribution of a particular profile will have a limited impact on the output of a computation. More precisely, differential privacy is formally defined as follows (Dwork et al., 2014).

**Definition 1** (Differential privacy). *A randomized algorithm $\mathcal{M}$ with domain $\mathbb{N}^{|\mathcal{X}|}$ is $\epsilon$-differentially private if for all $S \subseteq \mathcal{R}ange(\mathcal{M})$ and for all $x, y \in \mathbb{N}^{|\mathcal{X}|}$ such that $\|x - y\|_1 \leq 1$. $\Pr[\mathcal{M}(x) \in S] \leq e^\epsilon \cdot \Pr[\mathcal{M}(y) \in S]$.*

*Two datasets $x$ and $y$ are called* adjacent *(or* neighboring*) if they differ in the data of exactly one individual,* i.e., $\|x - y\|_1 \leq 1$.

In this paper, we will use the following mechanisms for building differentially-private counterfactuals. One way to satisfy differential privacy is through the *Laplace mechanism*, which adds Laplace noise to the output of the function. This Laplace noise is randomly sampled from a Laplace distribution defined based on the parameters related to the privacy budget and sensitivity of the function (Dwork et al., 2006). Specifically, the sensitivity $s$ represents the maximum change to the function output $f \in \mathcal{R}$ between two adjacent datasets that differ by one row. More formally, to ensure $\epsilon$-DP for a query $f(x)$, the Laplace mechanism is applied as follows: $\mathcal{M}(x) = f(x) + Lap\left(\frac{s}{\epsilon}\right)$.

The *exponential mechanism* (McSherry & Talwar, 2007) enforces differential privacy by selecting a response not solely based on accuracy, but by randomly choosing from a range of possible answers. The selection process is guided by probabilities assigned to each potential response, which are determined according to how well each response aligns with the function's objective. More formally, given a set $\mathcal{R}$ of possible outputs (solutions) and a scoring (utility) function $u : \mathcal{D} \times \mathcal{R} \to \mathbb{R}$ with sensitivity $\Delta u$, the exponential mechanism returns $r \in \mathcal{R}$ with a probability proportional to: $e^{\left(\frac{\epsilon u(x,r)}{2\Delta u}\right)}$.

*Report noisy max* (Dwork et al., 2014) is another mechanism selecting the best response to a query in a differentially-private manner. Consider, for instance, the scenario of returning the maximum count of some items, report noisy max works by adding independently generated Laplace noise $Lap(\frac{1}{\epsilon})$ to all counts and then selecting the maximum count among these noisy values. Finally, *randomized response* (RR) was introduced by Warner (1965) as a surveying technique guaranteeing the privacy of individuals when responding to a sensitive query. Randomized response flips the user's response based on some probability, while otherwise reporting the original answer. For instance, when reporting the value of a bit, its true value is reported with probability $p = \frac{e^\epsilon}{e^\epsilon+1}$ while being flipped with probability $1 - p$ (Kairouz et al., 2016). Randomized response is the basis for the implementation of local differential privacy (Dwork et al., 2014).

*(Sequential) composition theorem*: Let $M_1 : \mathbb{N}^{|X|} \to \mathcal{R}_1$ be an $\varepsilon_1$-differentially private algorithm, and let $M_2 : \mathbb{N}^{|X|} \to \mathcal{R}_2$ be an $\varepsilon_2$-differentially private algorithm. Then their combination, defined to be $M_{1,2} : \mathbb{N}^{|X|} \to \mathcal{R}_1 \times \mathcal{R}_2$ by the mapping:

$$M_{1,2}(x) = (M_1(x), M_2(x))$$

is $(\varepsilon_1 + \varepsilon_2)$-differentially private (Dwork et al., 2014).

## 3 Related Work

*Privacy attacks leveraging counterfactuals.* While counterfactuals, like other explanation techniques, help to gain users' trust by improving transparency, adversaries could exploit them to perform privacy attacks. In particular, model extraction attacks leveraging counterfactuals have been designed (Aïvodji et al., 2020; Kuppa & Le-Khac, 2021; Wang et al., 2022). More precisely, these previous works have shown that these attacks become more accurate when leveraging the counterfactual explanations compared to standard attacks that do not utilize them. Counterfactuals have also been used to perform membership inference attacks (Shokri et al., 2021; Kuppa & Le-Khac, 2021). Finally, more recently, Goethals et al. (2023) have introduced a new category of attacks called explanation linkage attacks, which work against instance-based counterfactual explanations. The attack assumes that the adversary will query the model for possible counterfactuals. Afterward, by receiving a counterfactual grounded in the training dataset, the adversary will use it to infer sensitive attributes belonging to an actual instance of the training dataset. While this attack is defined as an explanation linkage attack, it can be considered to implement an attribute inference attack (He et al., 2006) through linkage attacks leveraging counterfactuals. The same authors proposed a solution to mitigate this vulnerability using $k$-anonymity, guaranteeing that for each combination of feature values that can identify a record, there are at least $k-1$ instances sharing the same combination (Sweeney, 2002).

While $k$-anonymity improves the training dataset's privacy, it also has some drawbacks. First, altering the model's prediction is no longer guaranteed when providing $k$-anonymous counterfactuals. Second, from a privacy perspective, $k$-anonymity is known to be vulnerable to homogeneity attacks (Machanavajjhala et al., 2007). In practice, such attacks are possible when the sensitive attributes of all members in a group of $k$ records share the same value, in which case the adversary can infer this information without explicitly identifying the target record. Furthermore, $k$-anonymous explanations need to make specific assumptions about the adversarial knowledge and the dataset distribution, which makes them vulnerable to other privacy attacks (Cohen, 2022) like predicate singling-out (PSO) attacks (Altman et al., 2021).

*Differentially-private counterfactuals.* To mitigate the vulnerability of perturbation-based counterfactuals to membership inference and model extraction attacks, researchers have suggested adding differential privacy to the generation of counterfactuals. For instance, (Nelson, 2022; Pentyala et al., 2023; Ezzeddine, 2024) suggested using differential privacy mechanisms to generate synthetic data that can be used to produce counterfactuals. Meel et al. (2025) Suggested utilizing techniques from private information retrieval to generate private counterfactuals in the presence of limitations of changeable features.

More precisely, Nelson (2022) first train a black-box model on real data. Then, using various differentially-private techniques (DP-GAN, WDP-GAN, and DP-CTGAN), they generate differentially-private synthetic datasets, which Dice (the counterfactual generation algorithm they applied) uses as the training set to generate counterfactuals. They used this technique to mitigate the vulnerability to membership inference attacks. Pentyala et al. (2023) proposed a differentially-private recourse path based on the FACE (Poyiadzi et al., 2020) algorithm. More precisely, they suggested an end-to-end differential privacy pipeline to generate a recourse path. First, they trained a differentially-private black-box model on the real dataset using PATE (Papernot et al., 2018) or DP-SGD (Abadi et al., 2016). Then, a differentially-private clustering technique was applied to the training dataset, achieving cluster centers as differentially-private representatives of the clusters. To count the number of instances in each cluster, Laplace noise is added to the real counts, and these DP-values are used to evaluate the density while generating a recourse path.

Other works train differentially-private models and then generate counterfactuals for those models to protect against model extraction attacks. Mochaourab et al. (2022) trained an SVM classifier, then proposed adding a differentially-private demonstrator to the model to be used for counterfactual generation. Since this differentially-private version of the model loses accuracy and misclassifies some instances, only the robust counterfactuals that are classified in the counterfactual class by both models are returned to the user. Huang et al. (2023) suggested two methodologies to generate differentially-private counterfactuals: first, training a differentially private logistic regression model on real data and generate differentially private counterfactuals using this model, and second, applying post-processing differential privacy techniques, where Laplace noise is applied to the model's predictions, and a noisy counterfactual is generated that its differentially private prediction puts it in the counterfactual class. Finally, a recent work (Yang et al., 2022) relies on functional

mechanisms (Zhang et al., 2012) to incorporate differential privacy into perturbation-based counterfactual methods to simultaneously prevent membership inference and model extraction. They use an autoencoder to generate differentially-private class prototypes and return these prototype profiles as counterfactuals. A summarized overview of these differentially-private counterfactuals are presented in Table 1.

| | Target | Technique | Objective | Benefits | Limitations |
|---|---|---|---|---|---|
| Nelson (2022) | PB | DPS | MI prevention | MI prevention | No plausibility |
| Pentyala et al. (2023) | RP | DPS | Realistic recourse path | High robustness, MI prevention | Distribution change, No plausibility |
| Mochaourab et al. (2022) | PB | DPT | Robust DP explanation | Provide explanation | Model dependent |
| Huang et al. (2023) | PB | DPT, DPC | MI prevention | Accurate CF for big datasets | Model dependent, accuracy loss |
| Yang et al. (2022) | PB | DPC | MI and ME prevention | Robust counterfactual | No plausibility |

Table 1: Review of existing differentially private-mechanisms for counterfactual generation. CF is used as an abbreviation for counterfactuals, MI and ME refer, respectively, to membership inference and model extraction, PB stands for perturbation-based and RP for recourse path. Finally, DPS refers to differentially-private synthetic data, DPT to differentially-private training and DPC to differentially-private counterfactuals.

The main drawback of these methods is that they focus on generating differentially-private counterfactuals without having plausibility as a main objective in mind, which is one pivotal element of our suggested pipeline. Furthermore, some of them are model-specific. The main objective of these counterfactuals is to protect against membership inference and model extraction attacks. In contrast to these previous works, we propose differentially- private mechanisms for generating counterfactuals to protect the training data against re-identification attacks, which can be considered a form of linkage attacks. In addition, our approaches are model agnostic and can generate differentially-private counterfactuals that have high plausibility. For these different settings, comparing the utility and privacy aspects between these models and ours is not straightforward. Detailed explanations of each algorithm and their inconsistency to our setting is provided in the Appendix 1.

## 4 Differentially-Private Counterfactual Generation

In this section, we present our differentially-private approaches to generate privacy-preserving and plausible counterfactuals. More precisely, our framework is flexible in the sense that we have developed mechanisms for different stages of the counterfactual generation pipeline, which enables to control the privacy-utility trade-off in a fine-grained manner.

### 4.1 Pre-processed differentially-private counterfactuals

To apply differential privacy at the earliest stage of the pipeline, we propose to generate differentially-private dataset from real data(`DP-Pre`). More precisely, we first use real data to train our models. Then, to generate counterfactuals, instead of using the real training data, we input differentially-private datasets to the `NICE` mechanism to generate counterfactuals. In this differentially-private data generation, Laplace and randomized response mechanisms are applied to the training data, and then the obtained dataset is used to generate counterfactuals. This pre-processed differentially-private data generation takes the idea of applying Laplace noise and randomized response from local differential privacy (Dwork, 2006), while it is applied in the centralized setting and provides instance-level differential privacy guarantees. The idea of using local differential privacy techniques in a centralized setting has previously used to prevent explanation-guided privacy attacks (Nguyen et al., 2023). It should be taken into account that the model is trained on the original real data, and the differentially-private dataset is used only during the counterfactual generation mechanism. More precisely, we have created noisy versions of numerical features using the Laplace mechanism and applied randomized response to categorical features. The privacy budget assigned to each call of each Laplace mechanism and randomized response is $\frac{\epsilon}{d}$ in which $\epsilon$ is the overall privacy budget and $d$ is the dimensionality (*i.e.*, number of features). Considering the effect of the sequential composition (Dwork et al., 2006), this division maintains the privacy budget of $\epsilon$ for the solution.

## 4.2 In-processed differentially-private counterfactuals

Another mechanism we have developed to create differentially-private counterfactuals is called `Inline_DP`. The main idea behind this solution is that by increasing the number of participating instances in the counterfactual generation mechanism, new feature values are chosen from several instances instead of using only one instance, as in `NICE`. We select these neighbours following a scoring function, in which the score of each instance of the counterfactual class is computed based on Equation 1.

$$\text{score}_i = \max(\text{distance}) - \text{distance}_i \tag{1}$$

In a nutshell, this formula gives a higher score to instances closer to the original instance. Like `NICE` and our `DP-Pre` algorithm, we use the Heterogeneous Euclidean Overlap Method (HEOM) (Wilson & Martinez, 1997) with $L_1$ as our distance. Equation 2 defines this distance for two instances, $a$ and $b$, each having $d$ features.

$$\text{HEOM}(\mathbf{a}, \mathbf{b}) = \sum_{i=1}^{d} D_i(a_i, b_i), \tag{2}$$

in which $D_i(a_i, b_i)$ is defined for each attribute $i$ as:

$$D_i(a_i, b_i) = \begin{cases} \frac{|a_i - b_i|}{\text{range}_i}, & \text{if attribute } i \text{ is numerical} \\ 1, & \text{if attribute } i \text{ is categorical and } a_i \neq b_i \\ 0, & \text{if attribute } i \text{ is categorical and } a_i = b_i \end{cases} \tag{3}$$

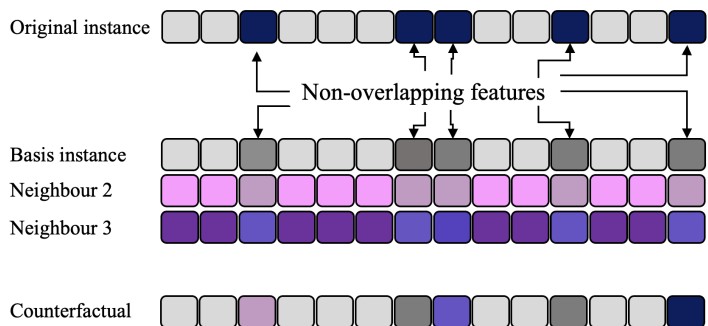

Figure 2: Illustration of `Inline_DP`. In this example, $k = 3$ nearest instances to the original instance are selected from the counterfactual class. Then, one instance is selected as the basis instance and non-overlapping features are identified. Using the Laplace mechanism and randomized response, non-overlapping values of all instances are privatized. At each iteration, the most rewarding feature value from all possible values among $k$ neighbours is selected and replaced in the original instance until the model changes its prediction.

After choosing the $k$ instances with the highest scores from the training dataset, one of these instances is randomly selected as the basis instance and non-overlapping features between this instance and the original instance requesting a counterfactual are identified. Then, for the features in the position of these non-overlapping values in all of $k$ neighbours, we apply the Laplace mechanism (for numerical features) and the randomized response (for categorical features). The fraction of the privacy budget assigned to each feature is $\frac{\epsilon}{k \times \#non\_overlapping}$ to maintain the global privacy budget of $\epsilon$. The sensitivity used for the Laplace noise is 1 for each feature value. After the privatization of feature values, a pool of these private values is created, and at each iteration, the most rewarding value is selected to replace the original instance. After choosing one value, all values for the position of the selected one are removed from the pool. This process continues until a counterfactual is generated or the pool is empty. This approach is illustrated in Figure 2 and described in more details in Algorithm 1.

In this algorithm, a basis instance is randomly selected to maintain the privacy budget. Another possibility is to use a fraction of the privacy budget to apply a differentially-private mechanism, such as the exponential mechanism (McSherry & Talwar, 2007), to choose the basis instance.

---

**Algorithm 1** In-processed differentially-private counterfactuals.

---

**Require:** x(query), ds(training_set), $k$(number of neighbours), $\epsilon$ (privacy budget)
**Ensure:** priv_cf (private counterfactual)
1: priv_cf ← x
2: Neighbours = knn(x,ds,k)
3: basis = sample(Neighbours)
4: $\Delta F = diff(X, basis)$ % find non-overlapping features
5: $\epsilon_{feature} = \frac{\epsilon}{\|\Delta F\|}$
6: diffVals ← vals($\Delta F$,neighbours) % values for non-overlapping features in all neighbours
7: privVals ← privatize(diffVals,$\epsilon_{feature}$) % lines 8-14 Algorithm 2
8: **while** priv_cf is not counterfactual **do**
9:    idx,val = pickBest(privVals) %choose best feature value from list;
10:    priv_cf$_{idx}$ ← val;
11:    update diffVals; %remove all values for feature index
12: **end while**
13: **return** priv_cf

---

### 4.3 Post-processed differentially-private counterfactuals

With respect to post-processing, we have designed three techniques to achieve differential privacy on counterfactuals after their creation (`DP-Post`), which we describe hereafter.

*Laplace mechanism and randomized response.* To make the counterfactuals generated by `NICE` differentially private, we applied the Laplace mechanism to numerical features and randomized response to categorical features. This approach operates *a posteriori*, meaning that it is applied after generating counterfactuals using `NICE`. Algorithm 2 illustrates the procedure for incorporating randomized response and the Laplace mechanism into `NICE` counterfactuals to achieve differential privacy. In this implementation, noise is added only to the feature values altered by `NICE`, with an upper bound on the number of modified features set to $d$. To ensure that the overall privacy budget remains within $\epsilon$, the privacy budget is divided equally among all modified features. According to the sequential composition theorem in differential privacy (Dwork et al., 2014), the cumulative privacy loss across multiple independent applications of a differential privacy mechanism is additive. Therefore, for $d$ modified features, the privacy budget allocated to each feature is $\frac{\epsilon}{d}$, ensuring the total budget does not exceed $\epsilon$.

---

**Algorithm 2** Post-processed differentially-private counterfactual generation

---

**Require:** $x$(query), $CF$ (nonprivat CF), $\epsilon$, $featVals$(feature values), $catfeats$(categorical features list)
**Ensure:** priv_cf (private counterfactual)
1: priv_cf ← $CF$
2: diffVals = diff($x$,priv_c)
3: $\epsilon_{feature} = \frac{\epsilon}{\|DiffVals\|}$
4: **for** $i \leftarrow 1$ to $lengh(X)$ **do**
6:    **if** $diffVals\_i = 1$ **then**
7:      **if** $i \in catfeat$ **then**
8:        $p = \frac{1}{1+\exp(\epsilon_{feature})}$
9:        with probability($p$):
10:        $priv\_cf_i \leftarrow$ sample($featVals$)
11:      **else**
12:        priv_cf$_i \leftarrow CF_i + lapNoise(scale = s/\epsilon_{feature})$ % s: sensitivity
13:      **end if**
14:    **end if**
15: **end for**
16: **return** priv_cf;

---

*Report Noisy Max.* In the first implementation of post-processing differential privacy techniques, the data distribution was not considered when choosing new feature values for generated counterfactuals. More precisely, when applying Laplacian noise and randomized response, the only information taken into account about the data distribution was the range of values for each feature. Therefore, applying these differentially-private mechanisms to generate counterfactuals results in a loss in plausibility and correctness. This means that changes in the feature values are significant enough to move counterfactuals to the undesired class. In

addition, even when they remain counterfactual, their distance might be quite far from query instances, thus making them less desirable from the user's point of view. To address this, we rely on another differentially-private mechanism, report noisy max, to engage the properties of the training dataset in updating generated counterfactuals. Similarly to other differentially private mechanisms, the privacy budget is divided by the number of changed features in the counterfactual class, and each of those features is updated using the noisy max value of the feature in the dataset. More precisely, the frequency of all existing values in the training data belonging to the counterfactual class is computed before the addition of Gaussian noise to this frequency. Afterwards, among these noisy values, the highest one is chosen, and its associated feature value is replaced in the counterfactual.

*Feature-based exponential mechanism.* The last approach that we have tried is the application of the exponential mechanism in the final stage of the pipeline. This approach also works at the feature level after receiving a non-private counterfactual generated by `NICE`. More precisely, similarly to Report Noisy Max, the frequency of each value repeated in the instances in the counterfactual class is measured, which plays the role of the utility function for the exponential mechanism. Each value receives a score based on these frequencies and an exponential mechanism is applied to choose a value based on them. The more frequent each value is in the counterfactual class of the training dataset, the higher score it receives as utility. Like other post-processing mechanisms, the privacy budget is divided between the number of features to keep the overall budget lower than $\epsilon$ following sequential mechanism principles. By doing so, the statistics of the data distribution in the counterfactual class will help to reduce the utility loss of the generated counterfactuals.

In the next section, we proceed to the empirical results of each algorithm.

## 5 Experimental Evaluation

In this section, we evaluate the approaches that we have developed for generating differentially-private counterfactuals. More precisely, we first describe the experimental setting before reporting on the results obtained.

### 5.1 Experimental Setting

The code and datasets to reproduce our results are available as supplementary materials.

*Datasets and models.* We have evaluated our differential privacy counterfactual generation methods on six datasets commonly used in the literature: `ACS income` (Ding et al., 2021) `adult` (Asuncion & Newman, 2007), `compas` (Angwin et al., 2016), `heloc` (OpenML, 2018), `ACS public coverage` (Ruggles et al., 2021) and `default credit` (Yeh & Lien, 2009). These datasets are all in the category of tabular data, which is the typical use case considered for counterfactual generation. In addition, all these datasets include sensitive information about individuals' financial, social or health status, as well as quasi-identifiers that adversaries can use to perform linking attacks that we detail hereafter.

Given the model-agnostic nature of our differentially-private counterfactual methods, we evaluated their performance across a diverse range of machine learning models: Random Forest (RF), Neural Networks (NN) and Light Gradient Boosting Model (LightGBM). For all these models, we have relied on their implementation on Scikit-Learn (Pedregosa et al., 2011). Our experiments were conducted on a cluster of 100 CPU nodes, each with 6GB of memory. Finally, the hyperparameter tuning was conducted via grid search, leveraging Scikit-Learn's model selection utilities to identify the optimal configurations for each combination of model and dataset. The final hyperparameter settings are detailed in Appendix C.

*Data preprocessing and training .* We have followed a unified approach for all datasets and model types to train our models. Each of the datasets was split into a training set that contains 70% of the data, test set (20%) and counterfactual set (10%) to ensure that the instances queried for counterfactuals have not been seen by the model before. Before training the model, we analyze the datasets to extract the required information for our counterfactual mechanisms. In particular, we distinguish between categorical and numerical features and establish the range of their values. The reported results are averaged over five executions of each algorithm, with models trained using five different random seeds, evaluated on 1000

instances under the settings described below. To account for the randomness introduced by differential privacy, each differentially-private mechanism was executed 20 times per instance.

The `NICE` library was used as the basis to generate differentially-private counterfactuals. For each model and dataset combination, we have chosen $\epsilon \in \{0.01, 0.1, 1, 5, 10\}$. In addition, to evaluate how the number of neighbours affects the quality of `Inline_DP` counterfactuals, we have run all the experiments of the exponential mechanism using four different values of the number of neighbours: $k \in \{3, 5, 10, 20\}$. The same setting was also used for all datasets, models and values of $\epsilon$ and $k$.

*Evaluation metrics.* To compare the performance of our methods, we rely on the following metrics.

- The *re-identification rate* quantifies how many exact matches each generated counterfactual has in the training dataset. This metric is classically used in the literature to measure the effectiveness of linkage attacks (Herzog et al., 2007; Vidanage et al., 2022). The rationale here is that the generated counterfactual is considered robust against re-identification if it has no exact match in the training dataset or if there are many matches, thus making it hard for an adversary to identify a specific profile. In contrast, the worst-case situation occurs when only one match exists in the training data, which means that the generated counterfactual discloses sensitive data. We use $M_0$ (respectively $M_1$) to denote the proportion of counterfactual explanations having no match (respectively exactly one match) in the training dataset.

$$M_0 = \frac{1}{|X_{CF}|} \sum_{x \in X_{CF}} \mathbb{I}(x \notin X_{train}). \tag{4}$$

$$M_1 = \frac{1}{|X_{CF}|} \sum_{x \in X_{CF}} \mathbb{I}((\sum_{x' \in X_{train}} \mathbb{I}\{x' = x\}) = 1). \tag{5}$$

- The *approximate match($d_{min}$)* corresponds to the minimum distance to the instances in the training set. This measure identifies the closest instance in the training dataset to the generated counterfactual and serves as a proxy for an approximate match. A smaller $d_{min}$ indicates a higher privacy risk, as the generated counterfactual could potentially reveal information about existing instances in the training set. Following the literature Brughmans et al. (2023), HEOM distance (see Equation 2) has been used as the measure for this purpose.

- The *recourse cost* (Wachter et al., 2017; Karimi et al., 2020; 2022) quantifies the difficulty for a user to change the decision made by the model by computing the distance between the original instance and its counterfactual. More precisely, following NICE, we choose HEOM (Wilson & Martinez, 1997) using the $L_1$ norm as our recourse cost. We use $Dist_{rec}$ to denote the average recourse cost computed over all generated counterfactuals.

- The *plausibility* assesses the realistic aspect of counterfactuals according to the training data distribution. While there is a huge literature on possible measures of plausibility, in this work we used the average distance to $k$-nearest neighbours in the training set as suggested in previous works (Dandl et al., 2020; 2024) to assess the generated counterfactuals. One advantage of using this measure is that the effect of potential outliers on the measurements is minimized due to the use of training instances.

- Finally, we define the *validity*, hereafter referred to as `Correctness`, as the proportion of counterfactual explanations that are correct (*i.e.*, whose predicted outcomes differ from the original instance).

## 5.2 Experimental Results

We conducted experiments to validate our differentially-private counterfactual generation mechanisms across all datasets and models described in the previous section. The results tables present an evaluation of the counterfactuals with respect to the metrics defined in Section 5.1, in which $Dist_{rec}$, $Plaus_{prox}$, and

*Correctness* are utility metrics, and $M_0$, $M_1$ and $d_{min}$ are privacy metrics. Notably, similar trends were observed across the other models and datasets, as detailed in the supplementary materials. This regular trend has been seen for all datasets, except for the ones with a higher number of features (`default credit`, `heloc`and `ACS public coverage`) for some settings, which are explained in the supplementary materials. For these datasets, the privacy metrics maintain high values due to the increased number of features, which leads to decreasing re-identification rates.

**Pre-processed differentially-private counterfactuals.** As described in Section 4.1, our initial approach to incorporating privacy into our counterfactuals involved generating a differentially-private dataset and subsequently using this dataset to produce counterfactuals. Table 2 illustrates the privacy-utility trade-offs of using differentially-private perturbed data via the Laplace mechanism and randomized response to produce counterfactuals over various privacy budgets. This mechanism achieves a high degree of privacy but significantly decreases utility. In contrast, increasing the privacy budget lessens the impact on utility. While almost all generated counterfactuals (more than 99.94%) have no match in the training dataset with more than 200% improvement in $d_{min}$, this high level of privacy comes with a utility loss of respectively 95% and 59% in $\text{Dist}_{\text{rec}}$ and $\text{Dist}_{\text{Plaus}}$. More precisely, the impact of differentially-private mechanisms on the training data distributions is detailed in Appendix A. Applying differential privacy techniques on the dataset used to generate counterfactuals shifts the distribution, and consequently, shifts the generated counterfactuals further from the training dataset, resulting in lower utility. All the results presented are based on the setting explained in Section 5.1 on the `ACS income` dataset and RF models. Results for other models and other datasets are provided in the appendix. To establish a differentially-private baseline, we implemented DP-

| Method | Epsilon | $Dist_{rec} \downarrow$ | $Plaus_{prox} \downarrow$ | Correctness $\uparrow$ | $M_0 \uparrow$ | $M_1 \downarrow$ | $d_{min} \uparrow$ |
|--------|---------|-------------------------|---------------------------|------------------------|----------------|------------------|--------------------|
| NICE | - | **0.415** $\pm$ 0.33 | **0.350** $\pm$ 0.28 | **100.000** | 85.280 | 14.400 | 0.123 |
| LDP | 0.010 | 1.040 $\pm$ 0.74 | 0.687 $\pm$ 0.49 | **100.000** | 99.940 | **0.040** | **0.510** |
|  | 0.100 | 1.020 $\pm$ 0.73 | 0.672 $\pm$ 0.49 | **100.000** | 99.940 | **0.040** | 0.494 |
|  | 1.000 | 0.963 $\pm$ 0.71 | 0.642 $\pm$ 0.47 | **100.000** | 99.920 | 0.060 | 0.466 |
|  | 5.000 | 0.872 $\pm$ 0.67 | 0.590 $\pm$ 0.43 | **100.000** | 99.940 | **0.040** | 0.423 |
|  | 10.000 | 0.808 $\pm$ 0.63 | 0.556 $\pm$ 0.41 | **100.000** | **99.960** | **0.040** | 0.393 |

Table 2: Privacy-utility trade-off for `LDP` for acs_income, RF model. Results are averaged over 1000 counterfactuals generated per model, for five models each trained using one different random seeds.

GAN Nelson (2022), a differentially-private synthetic data generation process. First, a differentially-private generative adversarial network is trained on the original dataset to produce synthetic data that satisfies differential privacy guarantees. This synthetic dataset is subsequently employed to generate counterfactual instances for the predictive model originally trained on the original data. Results presented in Table 3 show that this method outperforms our `LDP` mechanism in utility metrics, while evaluations of privacy metrics confirm that our preprocessing technique yields more private counterfactuals.

| Method | Epsilon | $Dist_{rec} \downarrow$ | $Plaus_{prox} \downarrow$ | Correctness $\uparrow$ | $M_0 \uparrow$ | $M_1 \downarrow$ | $d_{min} \uparrow$ |
|--------|---------|-------------------------|---------------------------|------------------------|----------------|------------------|--------------------|
| NICE | - | **0.415** $\pm$ 0.33 | 0.350 $\pm$ 0.28 | **100.000** | 85.280 | 14.400 | 0.123 |
| *DPGAN* | 0.010 | 0.714 $\pm$ 0.59 | **0.339** $\pm$ 0.28 | **100.000** | 99.900 | 0.080 | 0.213 |
|  | 0.100 | 0.800 $\pm$ 0.63 | 0.436 $\pm$ 0.3 | **100.000** | **99.920** | **0.060** | **0.294** |
|  | 1.000 | 0.856 $\pm$ 0.68 | 0.398 $\pm$ 0.33 | **100.000** | 99.880 | 0.080 | 0.268 |
|  | 5.000 | 0.780 $\pm$ 0.66 | 0.420 $\pm$ 0.29 | **100.000** | **99.920** | **0.060** | 0.277 |
|  | 10.000 | 0.839 $\pm$ 0.74 | 0.436 $\pm$ 0.36 | **100.000** | 99.840 | 0.120 | 0.292 |

Table 3: Privacy-utility trade-off for *DPGAN* for acs_income, RF model.

**In-processing differentially private counterfactuals.** To identify the neighbours of the original instance, as described in Section 4.2, we rely on the HEOM distance. The effect of the privacy budget and

number of neighbours is evaluated and shown in Table 4. It can be seen that this method significantly improves the privacy of `NICE` in terms of $M_0$, $M_1$ and $d_{min}$. In particular, the results show that $M_1$ decreases to less than 0.1 for all $\epsilon$ values. Also, for all combinations of $\epsilon$ and $k$, $M_0$ is higher than 99.82%. More precisely, $M_0$ and $M_1$ have been improved by more than 17% and 99%, respectively, and $d_{\min}$ increases by more than 100% for the least private setting. These results indicate that over 99% of the worst-case scenarios (as captured by $M_1$) have been mitigated, and more than 99.89% of the generated counterfactuals produced by this mechanism do not re-identify any instance from the training dataset. The increase of more than 100% in $d_{\min}$ further confirms the substantial reduction in vulnerability to approximate record-matching attacks. However, this improvement in privacy comes at a cost in terms of utility. Indeed, while increasing the privacy budget reduces the privacy guarantees as expected, it reduces the utility cost introduced by this algorithm. For $\epsilon = 10$ and $k = 20$, the lowest utility loss is achieved, which equals respectively 13%, 15% and 0.01% for $Dist_{rec}$, $Dist_{Plaus}$ and `Correctness`. In contrast, the greatest enhancement in privacy across different privacy budgets is obtained when counterfactuals are generated using 3 or 5 neighbours, whereas the utility metrics attain their highest values in configurations with a larger number of neighbours, in particular when 20 neighbours are employed. Another trend seen in the results is that increasing the size of neighbour sets used to generate counterfactuals improves utility while still achieving high privacy compared to smaller ones.

| Method | Epsilon | K | $Dist_{rec} \downarrow$ | $Plaus_{prox} \downarrow$ | Correctness $\uparrow$ | $M_0 \uparrow$ | $M_1 \downarrow$ | $d_{min} \uparrow$ |
|---|---|---|---|---|---|---|---|---|
| NICE | - | 1 | **0.415** $\pm$ 0.33 | **0.350** $\pm$ 0.28 | **100.000** | 85.280 | 14.400 | 0.123 |
| Inline_DP | 0.010 | 3 | 1.085 $\pm$ 0.73 | 0.679 $\pm$ 0.47 | 82.502 | **99.974** | 0.021 | 0.502 |
| | | 5 | 1.086 $\pm$ 0.74 | 0.675 $\pm$ 0.47 | 97.686 | 99.965 | **0.019** | 0.497 |
| | | 10 | 1.037 $\pm$ 0.69 | 0.650 $\pm$ 0.46 | 99.140 | 99.961 | 0.020 | 0.475 |
| | | 20 | 1.031 $\pm$ 0.69 | 0.645 $\pm$ 0.46 | 99.277 | 99.942 | 0.034 | 0.471 |
| | 0.100 | 3 | 1.083 $\pm$ 0.73 | 0.678 $\pm$ 0.47 | 82.617 | 99.970 | 0.022 | 0.501 |
| | | 5 | 1.079 $\pm$ 0.74 | 0.670 $\pm$ 0.47 | 97.874 | 99.955 | 0.032 | 0.494 |
| | | 10 | 1.021 $\pm$ 0.69 | 0.640 $\pm$ 0.45 | 99.278 | 99.949 | 0.028 | 0.467 |
| | | 20 | 0.998 $\pm$ 0.67 | 0.627 $\pm$ 0.45 | 99.484 | 99.934 | 0.042 | 0.455 |
| | 1.000 | 3 | 1.048 $\pm$ 0.72 | 0.657 $\pm$ 0.46 | 84.019 | 99.957 | 0.030 | 0.482 |
| | | 5 | 1.011 $\pm$ 0.71 | 0.630 $\pm$ 0.45 | 98.739 | 99.939 | 0.046 | 0.458 |
| | | 10 | 0.892 $\pm$ 0.63 | 0.572 $\pm$ 0.42 | 99.812 | 99.925 | 0.054 | 0.405 |
| | | 20 | 0.782 $\pm$ 0.57 | 0.519 $\pm$ 0.39 | 99.940 | 99.887 | 0.088 | 0.359 |
| | 5.000 | 3 | 0.918 $\pm$ 0.66 | 0.580 $\pm$ 0.42 | 86.579 | 99.927 | 0.058 | 0.411 |
| | | 5 | 0.815 $\pm$ 0.61 | 0.526 $\pm$ 0.39 | 99.711 | 99.892 | 0.087 | 0.365 |
| | | 10 | 0.642 $\pm$ 0.49 | 0.457 $\pm$ 0.35 | 99.989 | 99.845 | 0.126 | 0.304 |
| | | 20 | 0.531 $\pm$ 0.42 | 0.422 $\pm$ 0.33 | 99.991 | 99.832 | 0.136 | 0.274 |
| | 10.000 | 3 | 0.804 $\pm$ 0.6 | 0.522 $\pm$ 0.38 | 87.174 | 99.870 | 0.108 | 0.358 |
| | | 5 | 0.693 $\pm$ 0.52 | 0.469 $\pm$ 0.35 | 99.856 | 99.849 | 0.127 | 0.313 |
| | | 10 | 0.547 $\pm$ 0.42 | 0.422 $\pm$ 0.33 | 99.989 | 99.829 | 0.145 | 0.273 |
| | | 20 | 0.470 $\pm$ 0.38 | 0.405 $\pm$ 0.32 | 99.992 | 99.820 | 0.157 | 0.257 |

Table 4: Privacy-utility trade-off for `Inline_DP` for acs_income, RF model.

While in almost all settings the `Correctness` is higher than 97% (for $k$ higher than 5), there are situations ($k = 3$) in which `DP-In` fails to generate counterfactuals and falls to 82.50% for the smallest value of $\epsilon$. Since noise is added to the feature values of the selected neighbours in the pool, regardless of how this noise will affect their positions to the decision boundary, these values may not lead to good alternatives for the final counterfactual. For smaller numbers of instances, the number of available feature values in the pool decreases, which are noisier in the presence of small $\epsilon$. Consequently, in some cases `Inline_DP` fails to generate valid counterfactuals for the smallest $k$.

*Post-processed differentially-private counterfactuals.* Table 5 shows the results of implementing the Laplace mechanism and randomized response on counterfactuals generated by `NICE`. The results obtained demonstrate that while this post-processing approach highly improves the privacy of counterfactuals (*i.e.,* > 17%

percentage increase for $M_0$, $> 98\%$ percentage decrease for $M_1$ and $> 141\%$ percentage increase for $d_{min}$), it does not perform well in terms of utility. Indeed, both proximity and plausibility experience a significant decrease. For instance, the $Dist_{rec}$ increases by more than 38% when using Laplace noise and randomized response and the $Dist_{Plaus}$ by more than 32% for all epsilon budgets. Furthermore, this differentially-private post-processing mechanism fails to achieve counterfactuals for more than 64% of experiments. This occurs due to the random nature of the differential privacy mechanism introduced to the final results.

| Method | Epsilon | $Dist_{rec} \downarrow$ | $Plaus_{prox} \downarrow$ | Correctness $\uparrow$ | $M_0 \uparrow$ | $M_1 \downarrow$ | $d_{min} \uparrow$ |
|---|---|---|---|---|---|---|---|
| NICE | - | **0.415** $\pm$ 0.33 | **0.350** $\pm$ 0.28 | **100.000** | 85.280 | 14.400 | 0.123 |
| *Laplace_Noise_DP* | 0.010 | 1.028 $\pm$ 0.7 | 0.724 $\pm$ 0.45 | 30.655 | 99.921 | 0.046 | **0.533** |
| | 0.100 | 1.021 $\pm$ 0.71 | 0.718 $\pm$ 0.46 | 30.502 | **99.930** | **0.026** | 0.528 |
| | 1.000 | 0.931 $\pm$ 0.7 | 0.660 $\pm$ 0.44 | 31.240 | 99.901 | 0.061 | 0.475 |
| | 5.000 | 0.698 $\pm$ 0.6 | 0.528 $\pm$ 0.38 | 33.727 | 99.794 | 0.164 | 0.352 |
| | 10.000 | 0.571 $\pm$ 0.51 | 0.463 $\pm$ 0.34 | 35.951 | 99.795 | 0.159 | 0.288 |

Table 5: Privacy-utility trade-off for *Laplace_Noise_DP* for acs_income, RF model.

The results in Table 6 show how report noisy max affects the privacy and utility of generated counterfactuals. It shows that compared to Laplace noise and randomized response, report noisy max achieves less private results with lower utility loss in terms of $Dist_{rec}$, $Dist_{Plaus}$ and `Correctness`, in which the $Dist_{Plaus}$ improves for $\epsilon \geq 1$ while still achieving high privacy improvements. This means that for datasets with lower privacy considerations, report noisy max is a better option when the objective is to generate more practical model-agnostic private counterfactuals.

| Method | Epsilon | $Dist_{rec} \downarrow$ | $Plaus_{prox} \downarrow$ | Correctness $\uparrow$ | $M_0 \uparrow$ | $M_1 \downarrow$ | $d_{min} \uparrow$ |
|---|---|---|---|---|---|---|---|
| NICE | - | **0.415** $\pm$ 0.33 | 0.350 $\pm$ 0.28 | **100.000** | 85.280 | 14.400 | 0.123 |
| Noisy Max | 0.010 | 0.606 $\pm$ 0.51 | 0.430 $\pm$ 0.32 | 45.604 | **99.542** | **0.356** | **0.274** |
| | 0.100 | 0.593 $\pm$ 0.51 | 0.356 $\pm$ 0.29 | 62.474 | 99.184 | 0.681 | 0.211 |
| | 1.000 | 0.575 $\pm$ 0.46 | 0.312 $\pm$ 0.27 | 68.919 | 98.981 | 0.882 | 0.172 |
| | 5.000 | 0.661 $\pm$ 0.47 | 0.309 $\pm$ 0.27 | 87.537 | 98.181 | 1.582 | 0.166 |
| | 10.000 | 0.685 $\pm$ 0.49 | **0.304** $\pm$ 0.27 | 90.021 | 97.814 | 1.920 | 0.161 |

Table 6: Privacy-utility trade-off for `Noisy Max` for acs_income, RF model.

The feature-based exponential mechanism provides a high level of privacy, combined with a higher utility cost in terms of counterfactuals' correctness compared to `Laplace_Noise_DP`. For instance, the feature-based exponential mechanism leads to more than 28% loss in `Correctness` and 0.87% loss in $Dist_{rec}$ compared to `NICE` for the highest $\epsilon$, which is 10. The noisy max generated counterfactuals are more plausible (as low as 4% loss in $Dist_{Plaus}$ and 18% in terms of `Correctness`), it provides the least private counterfactuals among all post-processed techniques. Thus, with respect to privacy, the Noisy Max exponential mechanism is, on average, more successful than other `DP-Post` mechanisms.

| Method | Epsilon | $Dist_{rec} \downarrow$ | $Plaus_{prox} \downarrow$ | Correctness $\uparrow$ | $M_0 \uparrow$ | $M_1 \downarrow$ | $d_{min} \uparrow$ |
|---|---|---|---|---|---|---|---|
| NICE | - | **0.415** $\pm$ 0.33 | **0.350** $\pm$ 0.28 | **100.000** | 85.280 | 14.400 | 0.123 |
| Feature based exponential mechanism | 0.010 | 0.635 $\pm$ 0.45 | 0.489 $\pm$ 0.33 | 17.783 | 99.938 | 0.062 | 0.336 |
| | 0.100 | 0.635 $\pm$ 0.46 | 0.494 $\pm$ 0.34 | 17.989 | **99.961** | **0.034** | **0.340** |
| | 1.000 | 0.608 $\pm$ 0.45 | 0.482 $\pm$ 0.33 | 19.029 | 99.910 | 0.090 | 0.328 |
| | 5.000 | 0.474 $\pm$ 0.45 | 0.422 $\pm$ 0.32 | 26.761 | 99.892 | 0.108 | 0.268 |
| | 10.000 | 0.418 $\pm$ 0.43 | 0.395 $\pm$ 0.3 | 36.475 | 99.873 | 0.127 | 0.231 |

Table 7: Privacy-utility trade-off for `Feature based exponential mechanism` for acs_income, RF model.

These trends are observed roughly for all datasets except for Adult, which shows different behavior in terms of `Correctness`, in which increasing $\epsilon$ results in lower correctness, which is the opposite of other datasets and

is not expected within the context of differential privacy. This may be related to the properties of the Adult dataset. This affects the `NICE` generated counterfactuals to the training dataset and their position relative to the decision boundary. `NICE` counterfactuals generated for the Adult dataset are closer to the original instances (*i.e.*, they have smaller $\text{Dist}_{\text{rec}}$) compared to those generated for `ACS income`, which is the most similar dataset to Adult. According to the definition of counterfactuals, a smaller $\text{Dist}_{\text{rec}}$ indicates that the counterfactual lies closer to the decision boundary. Counterfactuals located near the decision boundary are more sensitive to perturbations. Therefore, adding noise increases the likelihood of misclassification, leading to a decrease in `Correctness`. The direction of Laplace noise and randomized response is not limited in this work, and analyzing it is out of the scope of this work. It could be a direction for future work to analyze whether it is possible to tailor this noise to move counterfactuals toward the desired class while maintaining privacy, and as a result, keep the correctness of the generated counterfactuals.

*Summary of results.* After analyzing various techniques for generating privacy-preserving counterfactuals, our study found that the pre-processing and in-processing mechanisms provide the best utility-privacy trade-off while post-processing techniques worsen the existing trade-off for `NICE`. While the $\text{Dist}_{\text{rec}}$ and $\text{Dist}_{\text{Plaus}}$ are a bit high, these techniques are almost always capable of generating valid counterfactuals with the highest values for privacy measures. Detailed comparison over utility and privacy metrics across all models and datasets is presented in the Appendix E

According to differential privacy standards and best practices, including NIST SP 800-26 (Near et al., 2023), it is often impossible to meet all privacy requirements and keep the best utility simultaneously. Hence, model providers and users should decide based on their privacy and utility requirements which method can best address their needs. We provide below an overview of how the implemented techniques perform with respect to each privacy and utility metrics:

- Proximity ($\text{Dist}_{\text{rec}}$). While `NICE` leads to the best proximity overall, the lowest proximity cost of suggested differentially-private counterfactuals belongs to `Noisy Max` techniques, which is one of our `DP-Post` mechanisms (Section 4.3). However, like other postprocessing techniques, this technique exhibits lower `Correctness` compared to `DP-In` and `DP-Pre`.

- Plausibility ($\text{Dist}_{\text{Plaus}}$). According to our experiments, the most plausible differentially-private counterfactuals are generated using the `Noisy Max` technique, followed by `Feature based exponential mechanism`. `Noisy Max` also works better than the two other postprocessing techniques in terms of `Correctness`, but it sub-performs againts other mechanisms such as `DP-In` and `DP-Pre` regarding `Correctness` and privacy measures. This can be explained by the nature of this technique, which gives a higher chance of achieving the more common feature values in the dataset, increasing the probability of matching with existing instances in the training dataset. It should be mentioned that even the lowest privacy rates achieved by this technique using the highest privacy budget are still high enough (for the `ACS income` dataset, it is higher than 97.81% for $\epsilon = 10$), which still makes them a reliable choice.

- Correctness (`Correctness`). While `NICE` is always successful in generating counterfactuals for all instances, this is not the case for all differentially-private counterfactuals. More precisely, `DP-Pre` technique is always capable of generating valid counterfactuals, while for other methods this is not the case. In particular, `DP-In` succeeds to generate valid counterfactuals for more than 98% instances for $K \geq 5$ while for smaller $K$s, it generates valid counterfactuals for more than 82.5% of instances. The only category of mechanisms that falls short in terms of `Correctness` are `DP-Post` techniques. Thus, if `Correctness` is the most important metric for users (which is often the case, as this is the primary objective of counterfactual generation), they should decide between `DP-Pre` and `DP-In` techniques.

- Re-identification rate ($M_1$ and $M_0$). `DP-In` is the algorithm that provides the highest privacy for all settings among our differentially-private counterfactuals. While this comes with a very small cost of `Correctness`, an acceptable cost of plausibility and a high cost in terms of proximity, it could be considered as the best achievable trade-off between utility and privacy.

- Approximate match ($d_{min}$). While `Laplace_Noise_DP` achieves the highest distance from the closest instance in the training dataset, `LDP` and `Inline_DP` exhibit values within less than a 10% deviation. Moreover, for small neighborhood sizes, `Inline_DP` achieves a greater distance than `Laplace_Noise_DP` for most values of $\epsilon$.

Table 8 provides the best choices among the differentially-private techniques suggested in the paper when a specific metric is the priority for the user. To find out about which method of each category is suggested for every requirement, and the performance of each individual method for each metric, visit the evaluation diagrams in Appendix E.

| Priority | trade-off | Model-agnostic | Privacy | Correctness | $Rec_{cost}$ | $Plaus_{prox}$ |
|---|---|---|---|---|---|---|
| Mechanism | DP-Pre,Inline_DP | DP-Pre, DP-Post | DP-Pre, DP-Pre,Inline_DP | DP-Pre | Inline_DP | Inline_DP |

Table 8: The most suitable methods suggested for various priorities. Using this table, the developers can decide which differentially private mechanism can address their special needs.

## 6 Conclusion

In this work, we have proposed a novel approach for enhancing the privacy of counterfactuals by designing differentially private solutions for generating instance-based counterfactuals that can be used at different stages of the counterfactual generation pipeline. Our experiments demonstrate that for various models trained on numerous real-world datasets, our `DP-In` mechanism can improve privacy with the lowest cost in terms of utility. More precisely, the higher the number of neighbours used, the more private the counterfactuals generated, but also the better the utility. Additionally, our experiments have shown that `DP-Post` (*i.e.*, *a posteriori* differential privacy through randomized response and Laplace mechanisms) did not provide a good utility-privacy trade-off for privacy-preserving counterfactual generation. In contrast, the `DP-Pre` mechanism provides acceptable trade-offs and could be an interesting approach to generate private counterfactuals.

Moreover, our `DP-Post` mechanisms achieve a worse trade-off between utility and privacy compared to the `DP-In` mechanism, making them mostly interesting in the situation in which the user needs a model-agnostic differential privacy solution for which they are willing to pay a high utility cost. In other words, while `DP-Pre` and `DP-Post` mechanisms cannot achieve results as good as the `DP-In` mechanism, their model-agnostic nature makes them good candidates for scenarios in which the counterfactual generation mechanism should not be altered, but rather its output is privatized. In contrast, `DP-In` has been designed specifically to generate differentially-private counterfactuals based on `NICE`, which leads to high performance in terms of utility and privacy but limits its applicability. This opens the avenue for further research to provide more cost-effective model-agnostic counterfactuals in terms of pre-processed and post-processed differentially-private mechanisms using other differential privacy mechanisms, resulting in less utility loss. Another direction for future work is to assess the practical effectiveness of our techniques in defending against other privacy attacks, such as membership inference and model extraction attacks. Since differential privacy has been widely used in defending against membership inference attacks, it is possible to explore whether our suggested differentially private framework can be utilized to defend against this type of attack.

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

# A  DP-perturbed distributions

To see the impacts of generating perturbed data over $\epsilon$ on the data distribution, Figures 3 compare the data distribution between the DP (DP dataset obtained using Laplace noise and RR) and the original data distribution for `ACS income`. These comparisons demonstrate that increasing the privacy budget leads to the synthetic data distribution becoming more similar to that of the original dataset. Distribution changes of other datasets under differential privacy follow the same trend and are provided in the supplementary materials.

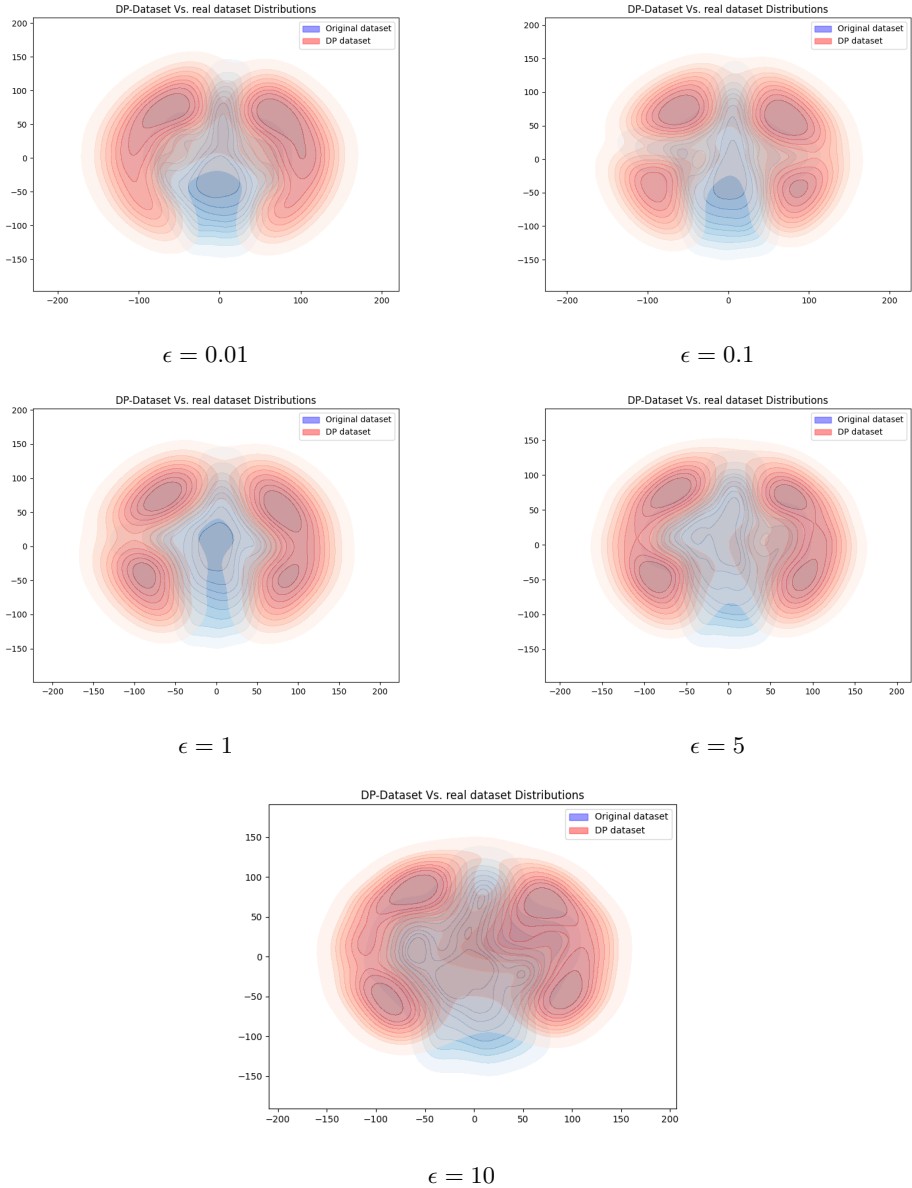

Figure 3: Distributions comparison of original and synthetic datasets for different values of $\epsilon$ for `ACS income` dataset.

# B    Datasets and their sensitive attributes

Table 9 summarizes the datasets used to perform the empirical study in this work. As shown in the table, all datasets are tabular with various numbers of attributes. Quasi-identifiers, sensitive attributes and class attributes are also defined in the table.

| Dataset | #attributes | Quasi identifiers | Sensitive Attribute | Class attribute |
|---|---|---|---|---|
| ACS income | 11 | Age, Sex, Race, Relationship, Marital status | Sex | Income |
| adult | 11 | Age, Sex, Race, Relationship, Marital status | Sex, Race | Income |
| compas | 9 | Sex, Age, Race | Sex , Race | Low_risk |
| default credit | 24 | Sex, Education, Mariage, Age | Sex | Default_payment |
| heloc | 24 | Credit history, utilization, inquiries | Credit history | RiskPerformance |
| ACS public coverage | 20 | Age, Sex, Race, Relationship, Marital status | Sex, Race, Nativity, DEAR, DEYE, DREM | Target |
| GiveMeSomeCredit | 11 | Age, MonthlyIncome, NumberOfDependents, | MonthlyIncome, NumberOfDependents | SeriousDlqin2yrs |

Table 9: Summary of the characteristics of datasets.

# C    Hyper-parameter settings for different models and datasets

We used GridSearchCV for hyperparameter tuning for all the models on all datasets to achieve highest accuracy of the models. Table 10 shows the optimum hyperparameters used in the model training phase of our experiments.

# D    Prediction accuracy of models

Table 11 represents the average accuracy of the models trained on our datasets. From each dataset, we excluded features that are either irrelevant for the decision-making process or redundant because they convey information already provided by other features. The obtained accuracy for all datasets is in the normal range of possible accuracy for each dataset. The low accuracy for `compas` and `heloc` dataset is due to the intrinsic difficult of the learning task, and in the literature Dressel & Farid (2018); Davis et al. (2022), most researchers achieved accuracy within this range.

| Dataset | NN | RF | LGBM |
|---|---|---|---|
| `ACS income` | activation: tanh
alpha: 0.01
hidden_layer_sizes: [50]
learning_rate: constant
solver: adam | max_depth: 20
min_samples_split: 10
n_estimators: 100 | colsample_bytree: 0.8
learning_rate: 0.1
max_depth: -1
min_child_samples: 20
n_estimators: 300
num_leaves: 50
reg_alpha: 1
subsample: 0.8 |
| `adult` | activation: tanh
alpha: 0.01
hidden_layer_sizes: [50]
learning_rate: constant
solver: adam | max_depth: 20
min_samples_split: 10
n_estimators: 100 | colsample_bytree: 0.8
learning_rate: 0.1
max_depth: 10
min_child_samples: 10
n_estimators: 100
num_leaves: 31
reg_alpha: 1
subsample: 0.8 |
| **Compas** | activation: relu
alpha: 0.01
hidden_layer_sizes: [50]
learning_rate: constant
solver: sgd | max_depth: 10
min_samples_split: 2
n_estimators: 100 | colsample_bytree: 1
learning_rate: 0.01
max_depth: 5
min_child_samples: 10
n_estimators: 100
num_leaves: 31
reg_alpha: 0
subsample: 0.8 |
| **Heloc** | activation: tanh
alpha: 0.01
hidden_layer_sizes: [50]
learning_rate: constant
solver: adam | max_depth: 20
min_samples_split: 10
n_estimators: 100 | colsample_bytree: 0.8
learning_rate: 0.1
max_depth: 5
min_child_samples: 20
n_estimators: 100
num_leaves: 31
reg_alpha: 1
subsample: 0.8 |
| `default credit` | activation: relu
alpha: 0.001
hidden_layer_sizes: [50,50]
learning_rate: constant
solver: sgd | max_depth: 10
min_samples_split: 2
n_estimators: 100 | colsample_bytree: 0.8
learning_rate: 0.01
max_depth: -1
min_child_samples: 10
n_estimators: 300
num_leaves: 100
reg_alpha: 0
subsample: 0.8 |
| `ACS public coverage` | activation: tanh
alpha: 0.01
hidden_layer_sizes: [50]
learning_rate: constant
solver: adam | max_depth: 20
min_samples_split: 10
n_estimators: 100 | colsample_bytree: 0.8
learning_rate: 0.1
max_depth: -1
min_child_samples: 10
n_estimators: 300
num_leaves: 50
reg_alpha: 1
subsample: 0.8 |

Table 10: Parameter settings for models across datasets.

|  | *NN* | *RF* | *LGBM* |
|---|---|---|---|
| *Adult* | 0.833 | 0.835 | 0.840 |
| `ACS income` | 0.810 | 0.810 | 0.829 |
| *Compas* | 0.679 | 0.677 | 0.678 |
| *def_cre* | 0.810 | 0.816 | 0.812 |
| *Heloc* | 0.720 | 0.720 | 0.729 |
| *ACS_pub_cov* | 0.809 | 0.811 | 0.813 |
| *GiveMeSomeCredit* | 0.934 | 0.9323 | 0.933 |

Table 11: Average model accuracy of different models trained on various datasets.

# E    Utility-privacy trade-offs over all models and datasets

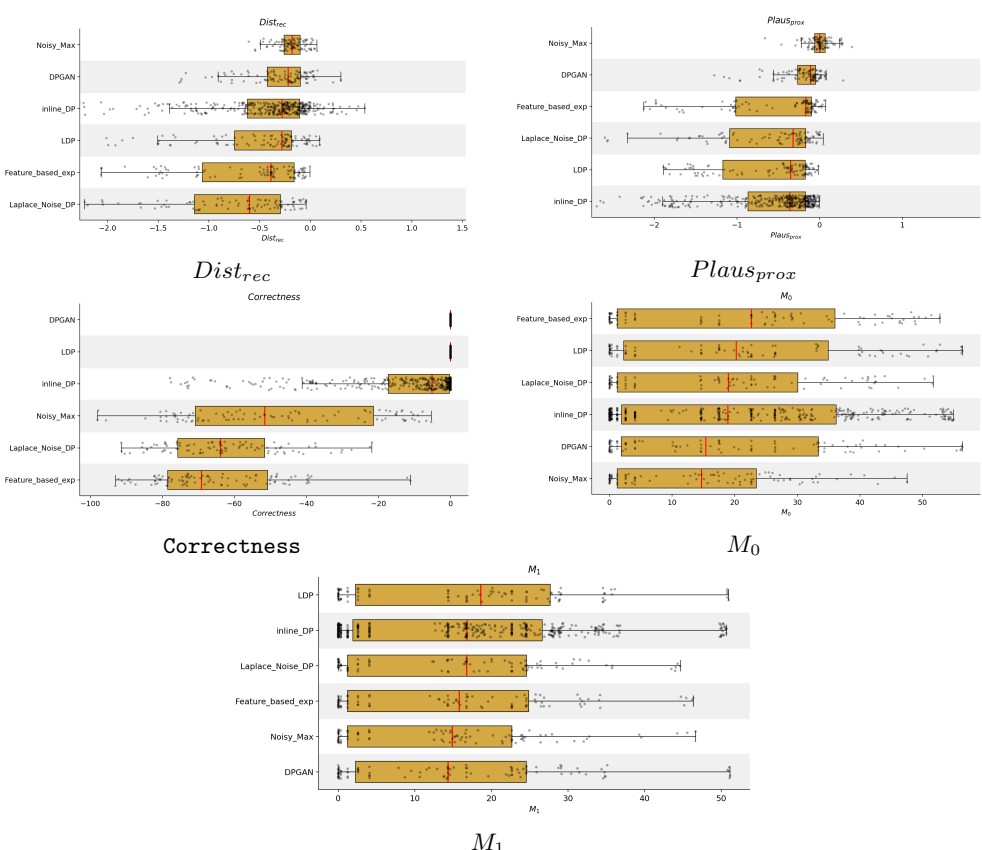

Figure 4: Improvements of all utility privacy measures of all DP techniques compared to `NICE`

# F    Existing DP mechanisms for Counterfactual generation

The following table explains why we did not implement other methods. The main reason is that these counterfactuals are mostly generated based on perturbation-based mechanisms, which, by design, are not vulnerable to counterfactual linkage attacks. Detailed explanations on the applicability of these methods to instance-based counterfactuals are provided in the table.

| Reference | Category | Summary and Implementation Notes |
|---|---|---|
| Nelson (2022) | DP synthetic data | DPGAN is implemented and the results are compared to our proposed techniques. |
| Pentyala et al. (2023) | DP counterfactual path | This research focuses on generating DP paths by generating DP-clusters of data where the DP paths go through the dense areas of the training dataset. To make these paths differentially private, cluster centers are selected using the exponential mechanism, where the center instance is chosen using the exponential mechanism over real instances. Then, noise is added to the number of instances at each cluster to make cluster densities differentially private, but in the end, the final counterfactual is not DP; only the path is DP. We tried to use the Laplace mechanism in cluster center selection, but it did not converge well, and utility measures were so poor that we did not include the results. |
| Mochaourab et al. (2022) | Robust explanation for DP model | This work does not provide DP counterfactuals. It implements a demonstrator for the model (an SVM model and its DP demonstrator), and robust counterfactuals are implemented, which are counterfactuals for both regular and DP models, but the counterfactuals are not differentially private. |
| Huang et al. (2023) | DP model training– DP prediction | Two implementations exist. First, training a DP logistic regression model and generating counterfactuals for it. This technique prevents model extraction attacks leveraging counterfactuals, but the generated counterfactuals are not differentially private; therefore, they are not comparable to DP counterfactuals. Second, a regular model is trained, and perturbation-based counterfactuals are generated for it. The technique used to apply DP to counterfactuals is adding noise to model predictions during the counterfactual generation process. For perturbation-based counterfactuals, this technique can improve privacy, but for NICE, which is the baseline method in our study, even with perturbed predictions, real feature values are used; therefore, this technique does not add privacy to the counterfactuals. |
| Yang et al. (2022) | Functional Mechanism | Designed for perturbation-based counterfactuals by generating DP class prototypes and adding a term including the distance to prototypes to the counterfactual generation optimization: $L_{CS}(\delta) = \alpha L_{pred} + \beta L_{dist} + \gamma L_{prot}$. This technique generates DP counterfactuals. This optimization does not apply to instance-based counterfactuals. Implementing this technique and comparing results would not provide a comparable baseline to our method. |

Table 12: Related work on differentially private synthetic data, counterfactual generation, and DP model training.

## G   Performances across all black-box Models and Datasets

In this section, results for `ACS income` on other models are presented. Since the results for the Adult dataset are different from those of other datasets, the results for Adult on the RF model are presented here, and the difference is explained here. The results for all models on other datasets are presented in the supplementary materials.

### G.1   `ACS income` - NN

| Method | Epsilon | $Dist_{rec} \downarrow$ | $Plaus_{prox} \downarrow$ | Correctness $\uparrow$ | $M_0 \uparrow$ | $M_1 \downarrow$ | $d_{min} \uparrow$ |
|--------|---------|------------------------|--------------------------|----------------------|---------|---------|---------|
| NICE | - | **0.43** $\pm$ 0.34 | **0.35** $\pm$ 0.29 | **100.00** | 82.40 | 16.82 | 0.11 |
| LDP | 0.01 | 0.81 $\pm$ 0.59 | 0.79 $\pm$ 0.55 | **100.00** | 99.90 | 0.06 | **0.61** |
| | 0.10 | 0.80 $\pm$ 0.59 | 0.78 $\pm$ 0.54 | **100.00** | **99.94** | 0.04 | 0.60 |
| | 1.00 | 0.75 $\pm$ 0.57 | 0.75 $\pm$ 0.53 | **100.00** | 99.90 | 0.06 | 0.57 |
| | 5.00 | 0.68 $\pm$ 0.52 | 0.68 $\pm$ 0.5 | **100.00** | 99.92 | **0.02** | 0.51 |
| | 10.00 | 0.61 $\pm$ 0.49 | 0.62 $\pm$ 0.47 | **100.00** | 99.92 | 0.08 | 0.45 |

Table 13: Privacy-utility trade-off for `LDP` for acs_income, NN model.

| Method | Epsilon | $Dist_{rec} \downarrow$ | $Plaus_{prox} \downarrow$ | Correctness $\uparrow$ | $M_0 \uparrow$ | $M_1 \downarrow$ | $d_{min} \uparrow$ |
|--------|---------|------------------------|--------------------------|----------------------|---------|---------|---------|
| NICE | - | **0.43** $\pm$ 0.34 | **0.35** $\pm$ 0.29 | **100.00** | 82.40 | 16.82 | 0.11 |
| DPGAN | 0.01 | 0.73 $\pm$ 0.64 | 0.38 $\pm$ 0.33 | **100.00** | 99.88 | 0.12 | 0.26 |
| | 0.10 | 0.62 $\pm$ 0.56 | 0.49 $\pm$ 0.35 | **100.00** | **99.98** | **0.02** | 0.34 |
| | 1.00 | 0.77 $\pm$ 0.64 | 0.48 $\pm$ 0.37 | **100.00** | 99.86 | 0.12 | 0.33 |
| | 5.00 | 0.67 $\pm$ 0.62 | 0.51 $\pm$ 0.37 | **100.00** | 99.92 | 0.06 | **0.35** |
| | 10.00 | 0.65 $\pm$ 0.6 | 0.43 $\pm$ 0.34 | **100.00** | 99.90 | 0.08 | 0.29 |

Table 14: Privacy-utility trade-off for $DPGAN$ for acs_income, NN model.

| Method | Epsilon | K | $Dist_{rec} \downarrow$ | $Plaus_{prox} \downarrow$ | Correctness $\uparrow$ | $M_0 \uparrow$ | $M_1 \downarrow$ | $d_{min} \uparrow$ |
|--------|---------|---|------------------------|--------------------------|----------------------|---------|---------|---------|
| NICE | - | 1 | **0.43** $\pm$ 0.34 | **0.35** $\pm$ 0.29 | **100.00** | 82.40 | 16.82 | 0.11 |
| Inline_DP | 0.01 | 3 | 0.92 $\pm$ 0.6 | 0.81 $\pm$ 0.53 | 93.63 | **99.97** | **0.02** | 0.62 |
| | | 5 | 0.86 $\pm$ 0.56 | 0.77 $\pm$ 0.51 | 99.39 | 99.95 | 0.03 | 0.58 |
| | | 10 | 0.82 $\pm$ 0.53 | 0.75 $\pm$ 0.51 | 99.71 | 99.94 | 0.04 | 0.56 |
| | | 20 | 0.82 $\pm$ 0.52 | 0.74 $\pm$ 0.51 | 99.77 | 99.94 | 0.03 | 0.56 |
| | 0.10 | 3 | 0.92 $\pm$ 0.59 | 0.81 $\pm$ 0.53 | 93.57 | 99.96 | 0.03 | 0.62 |
| | | 5 | 0.85 $\pm$ 0.55 | 0.76 $\pm$ 0.51 | 99.39 | 99.94 | 0.04 | 0.58 |
| | | 10 | 0.81 $\pm$ 0.52 | 0.74 $\pm$ 0.51 | 99.73 | 99.94 | 0.04 | 0.56 |
| | | 20 | 0.79 $\pm$ 0.52 | 0.73 $\pm$ 0.51 | 99.83 | 99.93 | 0.05 | 0.55 |
| | 1.00 | 3 | 0.89 $\pm$ 0.59 | 0.78 $\pm$ 0.52 | 94.07 | 99.94 | 0.05 | 0.59 |
| | | 5 | 0.80 $\pm$ 0.54 | 0.73 $\pm$ 0.51 | 99.72 | 99.92 | 0.06 | 0.54 |
| | | 10 | 0.71 $\pm$ 0.49 | 0.68 $\pm$ 0.5 | 99.92 | 99.89 | 0.07 | 0.50 |
| | | 20 | 0.63 $\pm$ 0.45 | 0.64 $\pm$ 0.49 | 99.97 | 99.85 | 0.12 | 0.46 |
| | 5.00 | 3 | 0.77 $\pm$ 0.55 | 0.68 $\pm$ 0.49 | 94.53 | 99.88 | 0.09 | 0.50 |
| | | 5 | 0.65 $\pm$ 0.47 | 0.63 $\pm$ 0.48 | 99.92 | 99.83 | 0.13 | 0.45 |
| | | 10 | 0.54 $\pm$ 0.4 | 0.58 $\pm$ 0.47 | 99.98 | 99.80 | 0.15 | 0.42 |
| | | 20 | 0.47 $\pm$ 0.36 | 0.56 $\pm$ 0.46 | 99.99 | 99.75 | 0.20 | 0.39 |
| | 10.00 | 3 | 0.68 $\pm$ 0.5 | 0.60 $\pm$ 0.45 | 94.20 | 99.83 | 0.13 | 0.42 |
| | | 5 | 0.57 $\pm$ 0.42 | 0.56 $\pm$ 0.44 | 99.95 | 99.79 | 0.16 | 0.40 |
| | | 10 | 0.47 $\pm$ 0.36 | 0.55 $\pm$ 0.45 | 99.98 | 99.77 | 0.20 | 0.38 |
| | | 20 | 0.43 $\pm$ 0.35 | 0.54 $\pm$ 0.45 | 99.99 | 99.71 | 0.24 | 0.38 |

Table 15: Privacy-utility trade-off for `Inline_DP` for acs_income, NN model.

| Method | Epsilon | $Dist_{rec} \downarrow$ | $Plaus_{prox} \downarrow$ | Correctness $\uparrow$ | $M_0 \uparrow$ | $M_1 \downarrow$ | $d_{min} \uparrow$ |
|---|---|---|---|---|---|---|---|
| NICE | - | **0.43** $\pm$ 0.34 | **0.35** $\pm$ 0.29 | **100.00** | 82.40 | 16.82 | 0.11 |
| *Laplace_Noise_DP* | 0.01 | 1.30 $\pm$ 0.85 | 0.96 $\pm$ 0.59 | 36.40 | 99.94 | 0.03 | **0.74** |
| | 0.10 | 1.29 $\pm$ 0.85 | 0.95 $\pm$ 0.59 | 36.64 | **99.94** | **0.03** | 0.74 |
| | 1.00 | 1.16 $\pm$ 0.84 | 0.86 $\pm$ 0.57 | 37.32 | 99.90 | 0.06 | 0.65 |
| | 5.00 | 0.82 $\pm$ 0.71 | 0.63 $\pm$ 0.47 | 39.71 | 99.77 | 0.17 | 0.44 |
| | 10.00 | 0.65 $\pm$ 0.59 | 0.52 $\pm$ 0.4 | 42.02 | 99.72 | 0.20 | 0.33 |

Table 16: Privacy-utility trade-off for *Laplace_Noise_DP* for acs_income, NN model.

| Method | Epsilon | $Dist_{rec} \downarrow$ | $Plaus_{prox} \downarrow$ | Correctness $\uparrow$ | $M_0 \uparrow$ | $M_1 \downarrow$ | $d_{min} \uparrow$ |
|---|---|---|---|---|---|---|---|
| NICE | - | **0.43** $\pm$ 0.34 | 0.35 $\pm$ 0.29 | **100.00** | 82.40 | 16.82 | 0.11 |
| Noisy Max | 0.01 | 0.72 $\pm$ 0.57 | 0.48 $\pm$ 0.36 | 48.26 | **99.68** | **0.22** | **0.32** |
| | 0.10 | 0.65 $\pm$ 0.55 | 0.34 $\pm$ 0.29 | 56.94 | 99.14 | 0.67 | 0.20 |
| | 1.00 | 0.61 $\pm$ 0.5 | 0.29 $\pm$ 0.26 | 60.31 | 98.93 | 0.86 | 0.15 |
| | 5.00 | 0.69 $\pm$ 0.51 | 0.28 $\pm$ 0.26 | 74.42 | 98.06 | 1.60 | 0.14 |
| | 10.00 | 0.74 $\pm$ 0.54 | **0.28** $\pm$ 0.26 | 77.69 | 97.47 | 2.17 | 0.14 |

Table 17: Privacy-utility trade-off for `Noisy Max` for acs_income, NN model.

| Method | Epsilon | $Dist_{rec} \downarrow$ | $Plaus_{prox} \downarrow$ | Correctness $\uparrow$ | $M_0 \uparrow$ | $M_1 \downarrow$ | $d_{min} \uparrow$ |
|---|---|---|---|---|---|---|---|
| NICE | - | **0.43** $\pm$ 0.34 | **0.35** $\pm$ 0.29 | **100.00** | 82.40 | 16.82 | 0.11 |
| Feature based exponential mechanism | 0.01 | 0.85 $\pm$ 0.61 | 0.65 $\pm$ 0.44 | 27.04 | 99.95 | 0.04 | 0.47 |
| | 0.10 | 0.85 $\pm$ 0.61 | 0.65 $\pm$ 0.44 | 27.04 | **99.96** | **0.04** | **0.47** |
| | 1.00 | 0.82 $\pm$ 0.61 | 0.63 $\pm$ 0.43 | 27.99 | 99.90 | 0.10 | 0.45 |
| | 5.00 | 0.64 $\pm$ 0.59 | 0.53 $\pm$ 0.41 | 36.37 | 99.91 | 0.09 | 0.35 |
| | 10.00 | 0.52 $\pm$ 0.54 | 0.46 $\pm$ 0.37 | 45.80 | 99.93 | 0.07 | 0.28 |

Table 18: Privacy-utility trade-off for `Feature based exponential mechanism` for acs_income, NN model.

### G.2 `ACS income`- **LightGBM**

| Method | Epsilon | $Dist_{rec} \downarrow$ | $Plaus_{prox} \downarrow$ | Correctness $\uparrow$ | $M_0 \uparrow$ | $M_1 \downarrow$ | $d_{min} \uparrow$ |
|---|---|---|---|---|---|---|---|
| NICE | - | **0.34** $\pm$ 0.3 | **0.37** $\pm$ 0.29 | **100.00** | 73.48 | 24.62 | 0.11 |
| LDP | 0.01 | 0.90 $\pm$ 0.67 | 0.81 $\pm$ 0.55 | **100.00** | 99.97 | 0.03 | **0.62** |
| | 0.10 | 0.88 $\pm$ 0.66 | 0.79 $\pm$ 0.56 | **100.00** | 99.93 | 0.07 | 0.61 |
| | 1.00 | 0.82 $\pm$ 0.63 | 0.76 $\pm$ 0.53 | **100.00** | 99.95 | 0.05 | 0.57 |
| | 5.00 | 0.71 $\pm$ 0.58 | 0.67 $\pm$ 0.49 | **100.00** | 99.97 | 0.03 | 0.49 |
| | 10.00 | 0.63 $\pm$ 0.54 | 0.61 $\pm$ 0.46 | **100.00** | **100.00** | **0.00** | 0.44 |

Table 19: Privacy-utility trade-off for `LDP` for acs_income, LGBM model.

| Method | Epsilon | $Dist_{rec} \downarrow$ | $Plaus_{prox} \downarrow$ | Correctness $\uparrow$ | $M_0 \uparrow$ | $M_1 \downarrow$ | $d_{min} \uparrow$ |
|---|---|---|---|---|---|---|---|
| NICE | - | **0.34** $\pm$ 0.3 | 0.37 $\pm$ 0.29 | **100.00** | 73.48 | 24.62 | 0.11 |
| DPGAN | 0.01 | 0.72 $\pm$ 0.63 | **0.35** $\pm$ 0.29 | **100.00** | 99.90 | 0.08 | 0.23 |
| | 0.10 | 0.70 $\pm$ 0.64 | 0.44 $\pm$ 0.33 | **100.00** | **99.95** | 0.05 | 0.30 |
| | 1.00 | 0.70 $\pm$ 0.65 | 0.44 $\pm$ 0.34 | **100.00** | **99.95** | **0.03** | 0.30 |
| | 5.00 | 0.70 $\pm$ 0.62 | 0.48 $\pm$ 0.34 | **100.00** | 99.93 | 0.05 | **0.33** |
| | 10.00 | 0.58 $\pm$ 0.51 | 0.42 $\pm$ 0.31 | **100.00** | 99.88 | 0.08 | 0.28 |

Table 20: Privacy-utility trade-off for `DPGAN` for acs_income, LGBM model.

| Method | Epsilon | K | $Dist_{rec} \downarrow$ | $Plaus_{prox} \downarrow$ | Correctness $\uparrow$ | $M_0 \uparrow$ | $M_1 \downarrow$ | $d_{min} \uparrow$ |
|---|---|---|---|---|---|---|---|---|
| NICE | - | 1 | **0.34** $\pm$ 0.3 | **0.37** $\pm$ 0.29 | **100.00** | 73.48 | 24.62 | 0.11 |
| Inline_DP | 0.01 | 3 | 0.98 $\pm$ 0.65 | 0.78 $\pm$ 0.53 | 89.22 | **99.95** | 0.04 | 0.59 |
| | | 5 | 0.95 $\pm$ 0.67 | 0.77 $\pm$ 0.53 | 97.68 | 99.94 | 0.05 | 0.58 |
| | | 10 | 0.93 $\pm$ 0.67 | 0.75 $\pm$ 0.53 | 98.85 | 99.94 | 0.05 | 0.57 |
| | | 20 | 0.92 $\pm$ 0.66 | 0.75 $\pm$ 0.53 | 99.08 | 99.94 | 0.05 | 0.56 |
| | 0.10 | 3 | 0.98 $\pm$ 0.65 | 0.78 $\pm$ 0.53 | 89.50 | 99.95 | **0.04** | 0.59 |
| | | 5 | 0.95 $\pm$ 0.67 | 0.77 $\pm$ 0.53 | 97.85 | 99.94 | 0.04 | 0.58 |
| | | 10 | 0.91 $\pm$ 0.66 | 0.74 $\pm$ 0.52 | 98.99 | 99.94 | 0.05 | 0.56 |
| | | 20 | 0.88 $\pm$ 0.65 | 0.73 $\pm$ 0.52 | 99.28 | 99.94 | 0.05 | 0.55 |
| | 1.00 | 3 | 0.94 $\pm$ 0.65 | 0.75 $\pm$ 0.52 | 90.48 | 99.92 | 0.07 | 0.56 |
| | | 5 | 0.88 $\pm$ 0.65 | 0.72 $\pm$ 0.52 | 98.88 | 99.92 | 0.07 | 0.54 |
| | | 10 | 0.77 $\pm$ 0.6 | 0.67 $\pm$ 0.5 | 99.75 | 99.90 | 0.08 | 0.49 |
| | | 20 | 0.66 $\pm$ 0.53 | 0.61 $\pm$ 0.47 | 99.89 | 99.88 | 0.10 | 0.44 |
| | 5.00 | 3 | 0.81 $\pm$ 0.6 | 0.66 $\pm$ 0.47 | 91.63 | 99.89 | 0.10 | 0.48 |
| | | 5 | 0.69 $\pm$ 0.55 | 0.61 $\pm$ 0.46 | 99.73 | 99.87 | 0.11 | 0.43 |
| | | 10 | 0.53 $\pm$ 0.44 | 0.55 $\pm$ 0.43 | 99.98 | 99.88 | 0.10 | 0.38 |
| | | 20 | 0.41 $\pm$ 0.36 | 0.52 $\pm$ 0.41 | 99.99 | 99.88 | 0.10 | 0.36 |
| | 10.00 | 3 | 0.70 $\pm$ 0.54 | 0.58 $\pm$ 0.43 | 90.91 | 99.90 | 0.09 | 0.41 |
| | | 5 | 0.57 $\pm$ 0.47 | 0.54 $\pm$ 0.41 | 99.83 | 99.87 | 0.12 | 0.37 |
| | | 10 | 0.43 $\pm$ 0.38 | 0.51 $\pm$ 0.41 | 99.98 | 99.87 | 0.12 | 0.35 |
| | | 20 | 0.35 $\pm$ 0.32 | 0.50 $\pm$ 0.4 | 99.99 | 99.89 | 0.10 | 0.34 |

Table 21: Privacy-utility trade-off for `Inline_DP` for acs_income, LGBM model.

| Method | Epsilon | $Dist_{rec} \downarrow$ | $Plaus_{prox} \downarrow$ | Correctness ↑ | $M_0$ ↑ | $M_1 \downarrow$ | $d_{min}$ ↑ |
|---|---|---|---|---|---|---|---|
| NICE | - | **0.34** ± 0.3 | **0.37** ± 0.29 | **100.00** | 73.48 | 24.62 | 0.11 |
| *Laplace_Noise_DP* | 0.01 | 1.26 ± 0.81 | 0.85 ± 0.56 | 37.77 | 99.95 | 0.05 | **0.65** |
| | 0.10 | 1.25 ± 0.82 | 0.85 ± 0.56 | 37.54 | **99.96** | **0.04** | 0.64 |
| | 1.00 | 1.13 ± 0.8 | 0.78 ± 0.54 | 36.58 | 99.92 | 0.07 | 0.58 |
| | 5.00 | 0.83 ± 0.71 | 0.61 ± 0.46 | 33.94 | 99.80 | 0.18 | 0.43 |
| | 10.00 | 0.63 ± 0.6 | 0.52 ± 0.4 | 33.26 | 99.76 | 0.20 | 0.34 |

Table 22: Privacy-utility trade-off for *Laplace_Noise_DP* for acs_income, LGBM model.

| Method | Epsilon | $Dist_{rec} \downarrow$ | $Plaus_{prox} \downarrow$ | Correctness ↑ | $M_0$ ↑ | $M_1 \downarrow$ | $d_{min}$ ↑ |
|---|---|---|---|---|---|---|---|
| NICE | - | **0.34** ± 0.3 | 0.37 ± 0.29 | **100.00** | 73.48 | 24.62 | 0.11 |
| Noisy Max | 0.01 | 0.80 ± 0.6 | 0.48 ± 0.35 | 41.55 | **99.78** | **0.21** | **0.33** |
| | 0.10 | 0.77 ± 0.59 | 0.38 ± 0.3 | 50.09 | 99.52 | 0.40 | 0.23 |
| | 1.00 | 0.72 ± 0.54 | 0.31 ± 0.26 | 56.03 | 99.17 | 0.73 | 0.18 |
| | 5.00 | 0.83 ± 0.52 | 0.28 ± 0.26 | 76.21 | 97.43 | 2.26 | 0.15 |
| | 10.00 | 0.88 ± 0.55 | **0.28** ± 0.26 | 79.58 | 96.47 | 3.23 | 0.15 |

Table 23: Privacy-utility trade-off for `Noisy Max` for acs_income, LGBM model.

| Method | Epsilon | $Dist_{rec} \downarrow$ | $Plaus_{prox} \downarrow$ | Correctness ↑ | $M_0$ ↑ | $M_1 \downarrow$ | $d_{min}$ ↑ |
|---|---|---|---|---|---|---|---|
| NICE | - | **0.34** ± 0.3 | **0.37** ± 0.29 | **100.00** | 73.48 | 24.62 | 0.11 |
| Feature based exponential mechanism | 0.01 | 0.80 ± 0.58 | 0.56 ± 0.4 | 26.94 | 99.96 | 0.04 | 0.40 |
| | 0.10 | 0.80 ± 0.58 | 0.56 ± 0.4 | 27.15 | 99.94 | 0.06 | **0.40** |
| | 1.00 | 0.79 ± 0.58 | 0.56 ± 0.39 | 27.51 | **99.96** | **0.04** | 0.39 |
| | 5.00 | 0.69 ± 0.58 | 0.52 ± 0.38 | 31.17 | 99.91 | 0.09 | 0.35 |
| | 10.00 | 0.51 ± 0.56 | 0.45 ± 0.36 | 40.57 | 99.92 | 0.08 | 0.28 |

Table 24: Privacy-utility trade-off for `Feature based exponential mechanism` for acs_income, LGBM model.

### G.3 `adult`- **RF**

| Method | Epsilon | $Dist_{rec} \downarrow$ | $Plaus_{prox} \downarrow$ | Correctness $\uparrow$ | $M_0 \uparrow$ | $M_1 \downarrow$ | $d_{min} \uparrow$ |
|---|---|---|---|---|---|---|---|
| NICE | - | **0.13** $\pm$ 0.11 | **0.19** $\pm$ 0.19 | **100.00** | 45.23 | 37.46 | 0.04 |
| LDP | 0.01 | 0.37 $\pm$ 0.21 | 0.36 $\pm$ 0.27 | **100.00** | **98.01** | **1.36** | 0.22 |
| | 0.10 | 0.38 $\pm$ 0.23 | 0.36 $\pm$ 0.27 | **100.00** | 97.53 | 1.82 | 0.22 |
| | 1.00 | 0.38 $\pm$ 0.26 | 0.36 $\pm$ 0.29 | **100.00** | 97.43 | 1.73 | 0.23 |
| | 5.00 | 0.38 $\pm$ 0.31 | 0.36 $\pm$ 0.31 | **100.00** | 96.13 | 2.65 | 0.22 |
| | 10.00 | 0.41 $\pm$ 0.38 | 0.38 $\pm$ 0.34 | **100.00** | 95.39 | 2.91 | **0.24** |

Table 25: Privacy-utility trade-off for `LDP` for adult, RF model.

| Method | Epsilon | K | $Dist_{rec} \downarrow$ | $Plaus_{prox} \downarrow$ | Correctness $\uparrow$ | $M_0 \uparrow$ | $M_1 \downarrow$ | $d_{min} \uparrow$ |
|---|---|---|---|---|---|---|---|---|
| NICE | - | 1 | **0.13** $\pm$ 0.11 | **0.19** $\pm$ 0.19 | **100.00** | 45.23 | 37.46 | 0.04 |
| Inline_DP | 0.01 | 3 | 0.46 $\pm$ 0.24 | 0.38 $\pm$ 0.26 | 86.58 | 98.63 | 0.89 | 0.24 |
| | | 5 | 0.40 $\pm$ 0.19 | 0.35 $\pm$ 0.25 | 90.52 | 98.83 | 0.74 | 0.22 |
| | | 10 | 0.38 $\pm$ 0.16 | 0.34 $\pm$ 0.24 | 92.88 | **98.89** | **0.66** | 0.21 |
| | | 20 | 0.38 $\pm$ 0.16 | 0.34 $\pm$ 0.24 | 94.62 | 98.73 | 0.77 | 0.21 |
| | 0.10 | 3 | 0.45 $\pm$ 0.24 | 0.37 $\pm$ 0.26 | 87.67 | 98.28 | 1.11 | 0.23 |
| | | 5 | 0.40 $\pm$ 0.19 | 0.35 $\pm$ 0.25 | 91.03 | 98.54 | 0.90 | 0.21 |
| | | 10 | 0.37 $\pm$ 0.16 | 0.33 $\pm$ 0.24 | 93.79 | 98.13 | 1.07 | 0.21 |
| | | 20 | 0.36 $\pm$ 0.16 | 0.32 $\pm$ 0.24 | 95.88 | 97.40 | 1.48 | 0.20 |
| | 1.00 | 3 | 0.42 $\pm$ 0.25 | 0.35 $\pm$ 0.26 | 92.36 | 96.37 | 2.05 | 0.22 |
| | | 5 | 0.35 $\pm$ 0.19 | 0.32 $\pm$ 0.24 | 95.41 | 95.60 | 2.50 | 0.19 |
| | | 10 | 0.30 $\pm$ 0.17 | 0.30 $\pm$ 0.24 | 97.57 | 93.88 | 3.47 | 0.17 |
| | | 20 | 0.25 $\pm$ 0.15 | 0.28 $\pm$ 0.23 | 99.06 | 91.51 | 4.84 | 0.16 |
| | 5.00 | 3 | 0.34 $\pm$ 0.24 | 0.31 $\pm$ 0.25 | 95.07 | 92.08 | 4.40 | 0.18 |
| | | 5 | 0.26 $\pm$ 0.18 | 0.28 $\pm$ 0.23 | 98.52 | 90.46 | 5.27 | 0.16 |
| | | 10 | 0.21 $\pm$ 0.14 | 0.26 $\pm$ 0.23 | 99.11 | 88.84 | 6.44 | 0.14 |
| | | 20 | 0.17 $\pm$ 0.12 | 0.25 $\pm$ 0.22 | 99.44 | 88.19 | 6.96 | 0.13 |
| | 10.00 | 3 | 0.28 $\pm$ 0.22 | 0.28 $\pm$ 0.24 | 93.14 | 89.40 | 5.67 | 0.16 |
| | | 5 | 0.23 $\pm$ 0.17 | 0.26 $\pm$ 0.23 | 98.46 | 88.05 | 6.43 | 0.14 |
| | | 10 | 0.18 $\pm$ 0.13 | 0.25 $\pm$ 0.22 | 98.86 | 87.15 | 7.18 | 0.13 |
| | | 20 | 0.15 $\pm$ 0.11 | 0.24 $\pm$ 0.22 | 99.27 | 87.31 | 7.36 | 0.13 |

Table 26: Privacy-utility trade-off for `Inline_DP` for adult, RF model.

| Method | Epsilon | $Dist_{rec} \downarrow$ | $Plaus_{prox} \downarrow$ | Correctness $\uparrow$ | $M_0 \uparrow$ | $M_1 \downarrow$ | $d_{min} \uparrow$ |
|---|---|---|---|---|---|---|---|
| NICE | - | **0.13** $\pm$ 0.11 | **0.19** $\pm$ 0.19 | **100.00** | 45.23 | 37.46 | 0.04 |
| *Laplace_Noise_DP* | 0.01 | 0.75 $\pm$ 0.44 | 0.48 $\pm$ 0.32 | 69.90 | **96.97** | **1.67** | **0.31** |
| | 0.10 | 0.74 $\pm$ 0.44 | 0.47 $\pm$ 0.32 | 69.68 | 96.87 | 1.73 | 0.31 |
| | 1.00 | 0.65 $\pm$ 0.43 | 0.44 $\pm$ 0.31 | 66.79 | 95.59 | 2.46 | 0.28 |
| | 5.00 | 0.43 $\pm$ 0.36 | 0.33 $\pm$ 0.27 | 60.72 | 91.37 | 4.76 | 0.19 |
| | 10.00 | 0.30 $\pm$ 0.27 | 0.28 $\pm$ 0.24 | 57.89 | 87.77 | 6.88 | 0.14 |

Table 27: Privacy-utility trade-off for *Laplace_Noise_DP* for adult, RF model.

As it can be seen in the results, the trend in correctness for `DP-Post`, when Laplace noise and randomized response are applied on `NICE` generated counterfactuals, is different from that of other datasets. Instead of increasing correctness with the privacy budget, higher privacy budgets for this dataset result in lower

| Method | Epsilon | $Dist_{rec}\downarrow$ | $Plaus_{prox}\downarrow$ | Correctness $\uparrow$ | $M_0\uparrow$ | $M_1\downarrow$ | $d_{min}\uparrow$ |
|---|---|---|---|---|---|---|---|
| NICE | - | $\mathbf{0.13}\pm 0.11$ | $0.19\pm 0.19$ | **100.00** | 45.23 | 37.46 | 0.04 |
| Noisy Max | 0.01 | $0.43\pm 0.32$ | $0.29\pm 0.25$ | 53.23 | **85.94** | **6.63** | **0.16** |
| | 0.10 | $0.32\pm 0.25$ | $0.20\pm 0.21$ | 58.00 | 70.67 | 12.50 | 0.10 |
| | 1.00 | $0.31\pm 0.22$ | $\mathbf{0.19}\pm 0.2$ | 76.15 | 72.18 | 13.58 | 0.08 |
| | 5.00 | $0.32\pm 0.22$ | $0.19\pm 0.2$ | 81.94 | 74.11 | 13.43 | 0.09 |
| | 10.00 | $0.32\pm 0.22$ | $0.19\pm 0.2$ | 82.10 | 74.07 | 13.42 | 0.09 |

Table 28: Privacy-utility trade-off for `Noisy Max` for adult, RF model.

| Method | Epsilon | $Dist_{rec}\downarrow$ | $Plaus_{prox}\downarrow$ | Correctness $\uparrow$ | $M_0\uparrow$ | $M_1\downarrow$ | $d_{min}\uparrow$ |
|---|---|---|---|---|---|---|---|
| NICE | - | $\mathbf{0.13}\pm 0.11$ | $\mathbf{0.19}\pm 0.19$ | **100.00** | 45.23 | 37.46 | 0.04 |
| Feature based exponential mechanism | 0.01 | $0.52\pm 0.34$ | $0.34\pm 0.26$ | 51.61 | 94.04 | 3.28 | **0.21** |
| | 0.10 | $0.51\pm 0.34$ | $0.34\pm 0.26$ | 51.82 | **94.12** | **3.25** | 0.20 |
| | 1.00 | $0.49\pm 0.34$ | $0.34\pm 0.26$ | 51.18 | 93.86 | 3.41 | 0.20 |
| | 5.00 | $0.33\pm 0.32$ | $0.29\pm 0.25$ | 51.28 | 91.63 | 4.80 | 0.15 |
| | 10.00 | $0.21\pm 0.23$ | $0.25\pm 0.23$ | 49.88 | 89.99 | 5.74 | 0.10 |

Table 29: Privacy-utility trade-off for `Feature based exponential mechanism` for adult, RF model.

correctness, which may be related to the properties of this dataset. This affects the `NICE` generated counterfactuals to the training dataset and their position relative to the decision boundary. The direction of Laplace noise and randomized response are not limited to this work, and analyzing it is out of the scope of this work. It could be a direction for future work to analyze if it is possible to conduct this noise to move counterfactuals toward the desired class while maintaining privacy, and as a result, keep the correctness of the generated counterfactuals.

### G.4 Other results

All the results for other datasets and models are provided in the supplementary materials. The source code of the project, which makes the results reproducible, is also provided.

