# OpenReview forum: "Differentially-private and plausible counterfactuals"
_TMLR — Rejected by TMLR_

### Review · Reviewer_CuzH · 2025-12-20

**Summary Of Contributions:**

- The manuscript proposes different methods to integrate differentiable privacy (DP) in order generate counterfactuals that are both plausible and robust to explanation linkage attacks.
- The proposed methods enable the integration of DP at different stages of the counterfactual generation pipeline, i.e. a) to produce DP data that is the used to generate counterfactuals (DP-Pre), b) within the counterfactual-generation process itself (Inline-DP), and c) on a post-hoc manner on the outputs produced by a conterfactual-generation method (DP-Post).

**Audience:**

Yes

**Audience Explanation:**

The ideas proposed in the manuscript have the potential of raising interest in a significant part of the TMLR audience. However, as indicated in Sec. 5.1, the empirical evaluation of the proposed method is exclusive on tabular datasets. As such, without additional evaluation showing the generalizability of the proposed method (and its performance) to other data modalities, the proposed method has a limited coverage of the TMLR audience.

**Broader Impact Concerns:**

To the best of my judgement, the contents of the manuscript do not have any ethical implication.

**Claims And Evidence:**

No

**Claims Explanation:**

Partly:
- Considering the stage at which the proposed method, the description of the proposed method (Sec. 4) and its empirical validation (Sec. 5) show that it is possible to have methods that can be integrated at different stages of the counterfactual generation pipeline and achieve some plausibility-privacy trade-off.
- On the general objective of the work, i.e. being capable of generating counterfactuals that are both plausible and provide privacy guarantees, as reported in the empirical evaluation, the level to which these two properties are guaranteed by different variants of the proposed method (DP-Pre, Inline-DP and DP-Post) is not that outspoken. Moreover, it is clear the level to which these properties are ensured depends on the variant being considered.

**Requested Changes:**

- The method describes the counterfactual-generation task in general terms. However, as admitted in Sec. 5.1, the empirical evaluation of the proposed method is exclusive on tabular datasets. While the counterfactual generation is a problem that has been traditionally studies in these type of datasets, there is a continuously growing number of efforts exploring other data modalities (see a selection below). Consequently, it would be more accurate  to indicate this focus on early parts of the manuscript.

  - Yash Goyal, Ziyan Wu, Jan Ernst, Dhruv Batra, Devi Parikh, Stefan Lee,  "Counterfactual Visual Explanations", International Conference on Machine Learning, PMLR 97:2376-2384, 2019.
  - Eoin Delaney, Arjun Pakrashi, Derek Greene, Mark T. Keane, “Counterfactual explanations for misclassified images: How human and machine explanations differ”, Artificial Intelligence, 2023.
  - Ashish Mishra, Gyanaranjan Nayak, Suparna Bhattacharya, Tarun Kumar, Arpit Shah, and Martin Foltin. 2024. LLM-Guided Counterfactual Data Generation for Fairer AI. ACM Web Conference 2024 (WWW '24).
  - Raphael Mazzine Barbosa de Oliveira, Kenneth Sörensen, David Martens, “A model-agnostic and data-independent tabu search algorithm to generate counterfactuals for tabular, image, and text data, European Journal of Operational Research, 2024.


- There are some parts of the manuscript with typos:
    - [pag-2] “… counterfactuals often sufferfrom low …” ([pdf](zotero://open-pdf/library/items/8N2WX3VG?page=2))
    - [pag-2] “This in-processed mechanism… ” ([pdf](zotero://open-pdf/library/items/8N2WX3VG?page=2))
    - [pag-6] “… provides instance-level DP guarantee.” ([pdf](zotero://open-pdf/library/items/8N2WX3VG?page=6))
    - [pag-10] “… supplementary materials.This regular trend …”
    - [pag-10] “… to generate counterfatuacle, shifts the …” ([pdf](zotero://open-pdf/library/items/8N2WX3VG?page=10))
    - [pag-14] “…to explore if our suggested algorithm can…” ([pdf](zotero://open-pdf/library/items/8N2WX3VG?page=14))

---

> ### Comment · Action_Editor_89ty · 2026-03-03
> **Reviewer Reminder: Submit your official recommendation for this paper**
>
> Hello Reviewer
>
> The authors have submitted their responses and updated the paper. This is a reminder to submit your official recommendation for this paper.

---

### Review · Reviewer_Pxog · 2025-12-26

**Summary Of Contributions:**

This paper proposes three methods in different stages of counterfactual generation to balance the trade-off between utility and privacy. They conducted intensive experiments to demonstrate the superiority of proposed algorithms.

**Audience:**

Yes

**Audience Explanation:**

This paper will appeal to researchers in counterfactual explanation as well as industry practitioners—such as those in banking—who seek to provide explainable services while preserving privacy.

**Broader Impact Concerns:**

No ethical concerns.

**Claims And Evidence:**

Yes

**Claims Explanation:**

1. This paper is well-written and motivated. It is easy to follow the storyline.
2. The research problem is interesting and appealing, which raises many concerns in counterfactual community.
3. The authors include comprehensive literature review to demonstrate the research background.
4. The authors follow the widely accept experiment settings and report abundant experiment results.

**Requested Changes:**

1. The related work section introduces several works that overcome privacy attack, or generate private counterfactual. is it possible to compare proposed methods with them?
2. Some typo errors. E.g., missing of space “sufferfrom” (summary of introduction), mode -> models, laplacien -> Laplacian (related work), which is -> which are (Experimental setting), Figures 3 to ?? (Appendix Section A).
3. Some abbreviations could be written as the full names, especially those appearing frequently in experiments.
4. It could be more clear to include dataset name, model name in Table captions.
5. The effective way is DP counterfactual is to increase the number of participant instances. However, the number K does not affect the experiment results greatly, why?
6. If we consider the privacy into generation, are there any changes in the while loop? Will it take more iterations?
7. Among three strategies, which one do you recommend? A table that discuss the most suitable scenarios of each could be preferred.

---

> ### Comment · Action_Editor_89ty · 2026-03-03
> **Reviewer Reminder: Submit your official recommendation for this paper**
>
> Hello Reviewer
>
> The authors have submitted their responses and updated the paper. This is a reminder to submit your official recommendation for this paper.

---

### Review · Reviewer_GFL3 · 2026-02-05

**Summary Of Contributions:**

The authors explore methods of providing privacy-preserving counterfactual explanations (CFEs) using differential privacy methods. Accordingly, the work analyses three methods of systematic noise addition to the counterfactual generation process, during three stages of generation: (i) Adding noise to the training data before using them in instance-based CFE generation; (ii) Adding noise to feature values during substitution; and (iii) Adding noise to the final CFE after the whole generation process. Empirical evaluations have been carried out on 6 datasets and 3 model types. However, the only baseline compared with is the NICE algorithm, which is also the base CFE method used in the newly proposed methods.

Strengths:
- The method addresses a problem which has implications to multiple verticals, not just limited to machine learning community.
- The proposed algorithms are easy to implement and intuitive.
- The presentation of the algorithms is clear and organized

Weaknesses:
- Experimental setup can be improved, in terms of both expanding the setup as well as reporting the results. Please see  the following comments for details.

**Audience:**

Yes

**Audience Explanation:**

Providing explanations such as CFEs for machine learning model outputs is a timely requirement, encouraged by both legal and societal expectations. Privacy is an integral part of it. Both of these aspects have been widely studied by the machine learning community, separately on their own as well as in conjunction. Hence, the problem this work focuses on is of interest for multiple audiences within the TMLR community.

**Broader Impact Concerns:**

No broader impact concerns.

**Claims And Evidence:**

No

**Claims Explanation:**

The study is based completely on empirical results. Current experiments span 6 real-world publicly available widely-used datasets and 3 common model types. While the authors have provided some details, the results need to be improved in both aspects of extending the experimental setup as well as proper reporting. Please see the below points:
- The selection of the datasets does not sufficiently cover all the possibilities given that even with the selected 6, one already deviates from the trend and there are no other datasets of the same kind to conclude the pattern.
- The authors do not provide a well-formed hypothesis, nor provide additional experiments exploring the reasons and the behavior of this deviation.
- The tables presenting the main results (Table 2, 3, 4, 5 and 6) do not properly detail the setup; are the reported numbers an average over all models and datasets? Or are they from just one model and/or dataset?
- While $M_0$ and $M_1$ (eq. 4 and 5) may be a good measure of privacy for ordinary instance-based CFE generation methods or even for just categorical features in the proposed methods, the suitability of these measures for numerical features where Laplace noise is being added is questionable as a good metric of privacy. This is due to the fact that even the slightest change to a single numerical feature will render $x’ \notin X_\text{train}$, but for an adversary, approximating the actual value within some accuracy might be sufficient. Therefore, can some normalized notion of closeness be integrated for evaluating the privacy of numerical features?
- The results in Appendix A Figure 3 is difficult to understand given that the axes have not been named or explained. Moreover, the overlap of the plots corresponding to original and DP datasets renders it even more difficult to identify the similarities/differences.
Given the above reasons, it is difficult to conclusively determine the gains offered by the proposed algorithms, and hence do not provide convincing and clear evidence.

**Requested Changes:**

Please see the comments above. Some additional suggestions:
- Although a comparison with the other DP methods (particularly the ones listed in Table 1) is not fair since they are not designed for instance’s privacy and plausibility, it is good to include them as baselines to see the gains of the proposed methods.
- Minor fixes:
  - Citations need formatting for when they are included in text vs cited independently (need to be within parenthesis)
  - The column $Plaus_{prox}$ in the result tables has not been introduced in the text — it should be $D_{plaus}$ as it seems.
  - Missing figure reference in Appendix A first line.
- Some related works to be included:
  - Meel, S., Dissanayake, P., Nomeir, M., Dutta, S., & Ulukus, S. (2025, June). Private Counterfactual Retrieval With Immutable Features. In 2025 IEEE International Symposium on Information Theory (ISIT) (pp. 1-6). IEEE.
  - Ezzeddine, F. (2024). Privacy implications of explainable AI in data-driven systems. arXiv preprint arXiv:2406.15789.
  - Hamer, J., Perello, N., Valladares, J., Viswanathan, V., & Zick, Y. (2024). Simple Steps to Success: A Method for Step-Based Counterfactual Explanations. Transactions on Machine Learning Research.
  - Dissanayake, P., & Dutta, S. (2024). Model reconstruction using counterfactual explanations: A perspective from polytope theory. Advances in Neural Information Processing Systems, 37, 83397-83429.

---

> ### Comment · Action_Editor_89ty · 2026-03-03
> **Reviewer Reminder: Submit your official recommendation for this paper**
>
> Hello Reviewer
>
> The authors have submitted their responses and updated the paper. This is a reminder to submit your official recommendation for this paper.

---

### Decision · Action_Editor_89ty · 2026-03-22

**Recommendation:** Reject

**Additional Comments:**

There are some formatting issues in a few places.

**Audience:**

Yes

**Audience Explanation:**

The topic is extremely interesting and timely, and can be very impactful for the AI/ML community and beyond.

**Claims And Evidence:**

No

**Claims Explanation:**

The paper addresses a very interesting problem at the intersection of privacy and explainability: providing counterfactual explanations using differential privacy methods. They study noise addition at three stages of generation (pre-, in-, and post-) and conduct empirical evaluations.
The topic and the problem formulation are extremely interesting and timely, as also noted by reviewers, and can be quite impactful for the ML community.

The reviewers acknowledge the edits in the updated version. Unfortunately, however, the reviewers feel the paper is still not ready for acceptance in its current form. One main concern (unanimous) was that the evidence is less convincing in terms of experiments, as mentioned by all three reviewers. Reviewers have noted that the strength of the experimental setup is quite limited, with inadequate baselines and counterfactual generation techniques (despite Appendix F), which might not be sufficient to support all the claims.

The authors are strongly encouraged to resubmit an updated version with a major revision that addresses all the comments and suggestions in greater detail.

**Resubmission Of Major Revision:**

The authors may consider submitting a major revision at a later time.